# The evolution of centriole degradation in mouse sperm

Sushil Khanal[1], Ankit Jaiswal[1], Rajanikanth Chowdanayaka[2], Nahshon Puente [1], Katerina Turner[1], Kebron Yeshitela Assefa[1], Mohamad Nawras [1], Ezekiel David Back [1], Abigail Royfman[1], James P. Burkett [3], Soon Hon Cheong [4], Heidi S. Fisher [5], Puneet Sindhwani[6], John Gray [1], Nallur Basappa Ramachandra [2] & Tomer Avidor-Reiss [1,6] ✉

Centrioles are subcellular organelles found at the cilia base with an evolutionarily conserved structure and a shock absorber-like function. In sperm, centrioles are found at the flagellum base and are essential for embryo development in basal animals. Yet, sperm centrioles have evolved diverse forms, sometimes acting like a transmission system, as in cattle, and sometimes becoming dispensable, as in house mice. How the essential sperm centriole evolved to become dispensable in some organisms is unclear. Here, we test the hypothesis that this transition occurred through a cascade of evolutionary changes to the proteins, structure, and function of sperm centrioles and was possibly driven by sperm competition. We found that the final steps in this cascade are associated with a change in the primary structure of the centriolar inner scaffold protein FAM161A in rodents. This information provides the first insight into the molecular mechanisms and adaptive evolution underlying a major evolutionary transition within the internal structure of the mammalian sperm neck.

Centrioles are subcellular organelles with an evolutionarily conserved, barrel-shaped structure that are found at the base of cilia and flagella[1,2]. There, the centriole, also known as the basal body, acts as a shock absorber that resists the movement of axonemal microtubules relative to one another, although they may flex in response to ciliary forces[3]. Yet, centrioles have evolved diverse structures in the sperm cells of many invertebrate and vertebrate animals, presumably due to divergent selective pressures produced by varying levels of sperm competition[4], an evolutionary process that drives sperm change due to post-ejaculatory competition between males[5–7].

Most non-mammalian vertebrates, such as fish and Tetrapods (e.g., turtles, snakes, lizards, and birds) have two canonical, barrel-like centrioles in their spermatozoa that are essential post-fertilization[8–12]. Also,

humans, cattle, and most other studied mammals have two spermatozoan centrioles, though only one (the proximal centriole) is structurally canonical. In these species, the other centriole–the distal centriole–has an atypical, fan-like structure and is part of a "transmission system" that connects the sperm tail to the head[13]. Both spermatozoan centrioles function post-fertilization to form the embryo's first two centrosomes, which organize the zygotic microtubule cytoskeleton[14–17]. In contrast, house mouse (*Mus musculus*) spermatozoa lack any recognizable centrioles, centrioles are absent in the early embryo, and viable offspring can be produced by injecting a sperm head that lacks centrioles into an oocyte[18–27]. How the centriole, an essential spermatozoan structure in most animals, became modified and, ultimately, dispensable in house mouse evolution is an enigma.

[1]Department of Biological Sciences, University of Toledo, Toledo, OH, USA. [2]Department of Studies in Genetics and Genomics, University of Mysore, Manasagangotri, Mysuru, India. [3]Department of Neurosciences, College of Medicine and Life Sciences, University of Toledo, Toledo, OH, USA. [4]Department of Clinical Sciences, College of Veterinary Medicine, Cornell University, Ithaca, NY, USA. [5]Department of Biology, University of Maryland College Park, College Park, MD, USA. [6]Department of Urology, College of Medicine and Life Sciences, University of Toledo, Toledo, OH, USA. ✉e-mail: Tomer.avidorreiss@utoledo.edu

One possible explanation is provided by the evolutionary sex cascade theory, which claims that sexual selection modifies gametes as logical consequences whenever certain successive conditions are met, forming a causality-connected events sequence[28]. This cascade is driven largely by post-ejaculatory selection, which includes sperm competition and cryptic female choice, to produce spermatozoa with varying centriole numbers, morphologies, and behaviors[7,29]. While the cascade theory is not testable, it has, up to now, explained mechanisms causing change in general gamete and gonad properties, such as anisogamy (gametes of different sizes)[30], sex ratio[31], and testis size[32]. Though there have been some attempts to connect sexual selection to the size of spermatozoa or the spermatozoan tail[33–36], how sexual selection applies in terms of major evolutionary transitions to an internal gamete structure, such as the centriole, remains less explored.

Synthesis of the literature on gametes and embryonic centrioles suggests a hypothetical, big stroke, four-stage, centriolar evolutionary cascade that has impacted the spermatozoa and embryos in mammals (Fig. 1a).

In Stage 1, the pre-mammalian centriolar configuration, spermatozoa include a canonical proximal centriole and a canonical distal centriole, representing the ancestral animal configuration. At this stage, the two centrioles are required for early embryogenesis

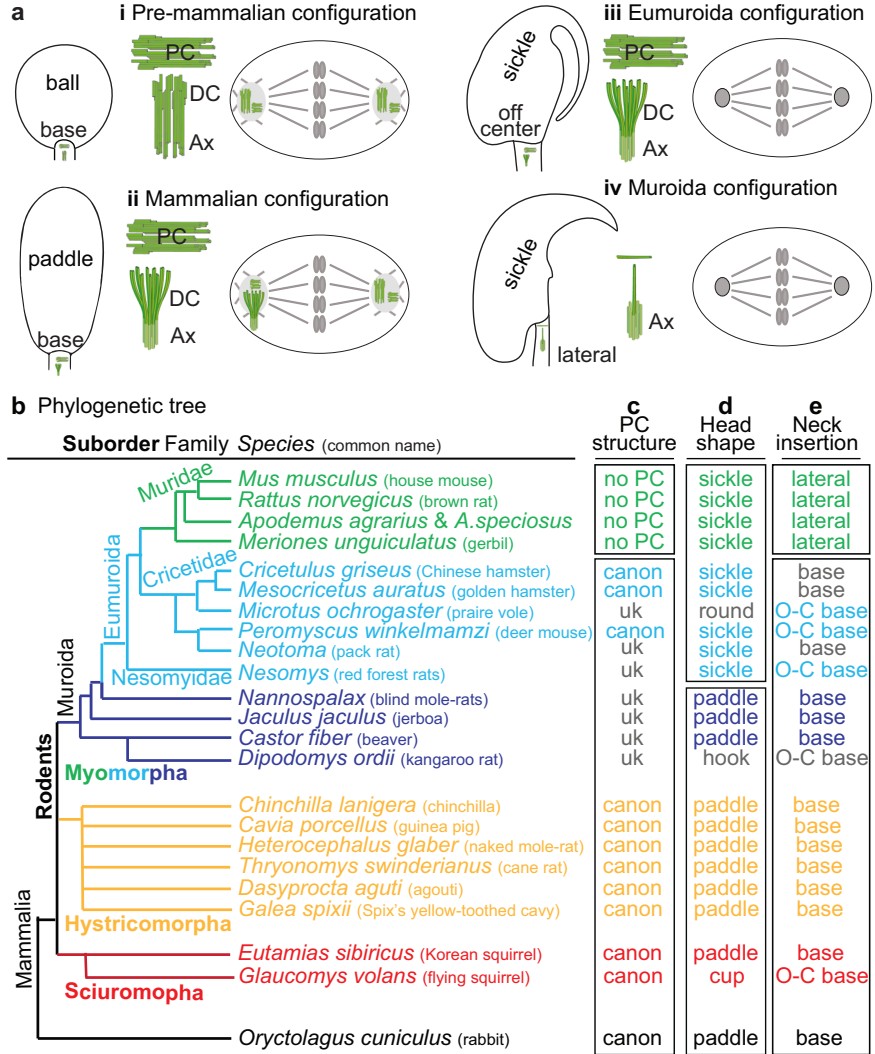

**Fig. 1 | Sperm proximal centriole degradation evolved after the separation of families Cricetidae and Muridae. a** The sperm centriole evolutionary cascade hypothesis. Each panel depicts the spermatozoan head and neck morphology (left), the spermatozoan centrioles (middle, in green), and the zygotic centriolar configuration (right). (**i**) Stage 1: the *pre-mammalian centriolar configuration*. Spermatozoon with a ball-shaped head, centrally inserted neck (tail attached below the head base center), and canonical proximal (PC) and distal centrioles (DC). Centriole-dependent zygote with two centrosomes emanating asters. (**ii**) Stage 2: the *mammalian centriolar configuration*. Sperm competition improved sperm behavior at this stage. Spermatozoon with a paddle-shaped head, centrally inserted neck, and canonical proximal and distal centrioles. Centriole-dependent zygote with two centrosomes emanating asters. (**iii**) Stage 3: the *Eumuroida* (subgroup of murids) *centriolar configuration*. The cost of increased miscarriage rates eliminated the need for zygotic centrioles at this stage. Spermatozoon with a sickle-shaped head, neck attached to the base either centrally or off-center (tail attached asymmetrically, below the head base), and canonical proximal and distal centrioles. Centriole-independent embryonic development. (**iv**) Stage 4: the *murid* (house mouse) *centriolar configuration*. Spermatozoan centrioles were freed from the functional constraints imposed by the centriole's role in the embryo, allowing for innovation in sperm morphology. Spermatozoon with a sickle-shaped head, lateral head-neck attachment (tail attached to the side of the head, parallel to the base), and remnant centrioles. Centriole-independent embryonic development. **b–e** A summary of rodent evolution, depicting their phylogenetic tree (**b**), proximal centriolar structure (**c**), head shape (**d**), and neck attachment (**e**). No PC, proximal centriolar structure not observed; uk, unknown; canon presence of a structurally canonical proximal centriole; sickle, sickle-shaped head; paddle, paddle-shaped head; lateral, neck attachment on one side of the head; O-C base, off-center neck attachment to the base of the head; base, neck attachment near the center of the base of the head.

(zygote to blastocyst), exhibiting centriole-dependent embryonic development[9].

In Stage 2, the mammalian centriolar configuration, sperm competition within the female reproductive tract of internal fertilizers selected for sperm with atypical centrioles. This suggests that atypical centrioles are an innovation associated with changes in the mode of fertilization[37–39], since internal fertilization provides unique challenges to the sperm. For example, in mammals, sperm swim in a multi-compartmentalized female reproductive tract with varying viscoelastic mucus and complex landscapes, including microgrooves[40]. Therefore, in mammals, atypical centrioles produced spermatozoa with a dynamic basal complex (DBC), an advantage that helps sperm to navigate the complex female reproductive tract[13]. At this evolutionary stage, centriole-dependent embryonic development is maintained. However, the atypical distal centriole comes with the costs of increased miscarriage rates and reduced fecundity[41].

In Stage 3, the Eumuroida (a murid subgroup that includes Muridae and Cricetidae) centriolar configuration, the miscarriage cost of the mammalian centriolar configuration resulted in reduced offspring production, acting as a selective pressure in the ancestor of house mice, an r-strategy species that maximizes reproductive capacity[42]. Consequently, centriole-independent early embryonic development (i.e., development of the embryo during the zygote and morula stages occurring without centrioles) evolved in the mouse ancestor[18,22,43–51]. Centriole-independent early embryonic development is mediated by an egg-derived mechanism involving maternal cytoplasmic centrosomes and post-cytokinetic microtubule bridges[47,52–55].

In Stage 4, the Muridae centriolar configuration, embryonic development became centriole-independent, and spermatozoan centrioles were freed from the functional constraints imposed by their role in the embryo. This freedom allowed the proximal and distal centriolar structures to be further modified, leaving only their remnants in spermatozoa (i.e., "degraded" or "degenerated"[24,56]) and enhancing spermatozoan competitiveness. In mice, this led to the evolution of a new type of sperm neck attachment at the side of the nucleus rather than at the base, as in most other animals.

General aspects of this sperm centriole evolutionary cascade hypothesis are supported by morphological and ultrastructural studies in the sperm, egg, and zygote; however, it is unclear how these changes occurred at the molecular level[18,22,43–51].

The canonical structure and function of centrioles are essential throughout mammalian development and physiology to form the cells' cilia[57]. Centrioles serve as both the seed and template for cilia, and therefore, their precise structure is critical for their function[57,58]. Considering this major constraint on centriolar structure, it is surprising that spermatozoan centrioles are so structurally diverse. Functional conservation on the one hand and structural diversity on the other produces genetic conflict and requires genomic plasticity, which should have a molecular signature of positive selection. Such plasticity should accrue in some centriolar proteins in some lineages and could be accompanied by gene duplication[59–62] or new protein isoforms, though how a remodeled sperm centriole is produced through centriolar plasticity is unclear.

One sub-cellular mechanism that contributes to sperm centriolar plasticity is the timing of their remodeling. Sperm centrioles are remodeled only after they form flagella (i.e., the sperm tail) in a process referred to as centrosome reduction[24]. In most mammals, centrosome reduction produces atypical centrioles and is therefore referred to as centriole remodeling in these species[63]; in mice, centrosome reduction eliminates the two sperm centrioles and is referred to as centriole degeneration or degradation[18]. Because centriole remodeling occurs in many mammalian orders, it is presumed to be the ancestral mammalian process; however, how centriole remodeling evolved into centriole degradation is unknown.

The rigid, barrel-shaped structure of canonical centrioles is maintained by an inner scaffold comprised of the proteins CETN1, POC5, POC1B, and FAM161A[64,65]. Interestingly, the sperm of many mammalian species–including humans, rabbits, and bovines–have evolved a fan-shaped distal centriole, in which the inner scaffold proteins are redistributed into two rod-like structures[63]. These rods, along with other surrounding structures (i.e., segmented columns and the proximal centriole), move relative to each other during tail beating, forming a dynamic basal complex that mechanically couples the sperm head and tail[13]. The fate of inner scaffold proteins in murid spermatozoa is unknown.

Here, we studied rodent sperm centrioles and found that centriole degeneration in rodents occurred only after the evolution of centriole-independent embryonic development in the ancestor of families Cricetidae and Muridae and was correlated with an evolutionarily novel, lateral, head-neck attachment in the spermatozoa of Muridae species. We found that the appearance and primary structure of an evolutionarily novel isoform of the inner scaffold protein FAM161A gradually evolved in the lineage leading to mice. The first changes to FAM161A were correlated with sperm centrioles becoming dispensable post-fertilization in the ancestor of families Cricetidae and Muridae, though the centrioles maintained their ancestral mammalian structure. Later, further changes to FAM161A were correlated with the appearance of an evolutionarily novel head-neck morphology and sperm centriole degeneration. Structural changes to FAM161A resulted in distinct functions in vitro and unique localization in vivo. Finally, although Muridae spermatozoa have no recognizable centrioles, centriolar proteins are present in their sperm neck, suggesting that highly modified remnant centrioles are present. Altogether, our study provides the first molecular evidence of an evolutionary process that we hypothesize changed the centriole remodeling program occurring in most mammals into the centriole degeneration program that occurs in some rodents.

## Results

### Proximal centriole loss and lateral head-neck attachment associatively evolved in the Muridae ancestor

The fan-shaped, atypical distal centriole was discovered in 2018 using advanced electron microscopy[63]; prior to this year, studies of spermatozoa mainly addressed the presence or lack of barrel-shaped centrioles. Two barrel-shaped, proximal and distal centrioles are found in the spermatozoa of most basal vertebrate external fertilizers[37]. In contrast, classic transmission electron microscopy (TEM) studies in mammalian spermatozoa found only one barrel-shaped centriole–the proximal centriole–with a few exceptions in marsupial and some rodent species that appear to lack it (Supplementary Fig. 1). Because rodent spermatozoan ultrastructure has undergone extensive phylogenetic study, and because the stereotypical proximal centriolar structure enables detection of evolutionary changes, we used rodent proximal centriolar ultrastructure to track the evolutionary changes leading to their disappearance. To this end, we systematically surveyed past TEM studies of rodent spermatozoan centrioles (Supplementary Fig. 1b, Supplementary Data 1).

The precise phylogeny of proximal centriole loss is unknown. In general, the literature on rodent sperm centrioles is confusing and at many times appears to claim that rodent spermatozoa lack a proximal centriole. For example, Manandhar and colleagues wrote that "rodent spermatozoa totally lack centrioles" p. 256[66]. Similarly, Xu and colleagues wrote that "spermatozoa of several species (e.g., rodents) lack centrioles due to complete centriolar degeneration" p. 201[67]. However, when we surveyed the original literature on rodent spermatozoan ultrastructure, we found that a proximal centriole was reported to be present in most rodents and in all three of the main rodent suborders (Supplementary Fig. 1b, Supplementary Data 1). A proximal centriole

was reported in all studied rodent species of suborder Hystricomorpha, which includes the guinea pig (*Cavia porcellus*)[68,69], chinchilla (*Chinchilla lanigera*)[70], cane rat (*Thryonomys swinderianus*)[71], agouti (*Dasyprocta aguti*)[72], Spix's yellow-toothed cavy (*Galea spixii*)[73], and naked mole rat (*Heterocephalus glaber*)[74]. A proximal centriole was also reported in the rodent species of suborder Sciuromorpha, which includes the Korean squirrel (*Tamias sibiricus*)[75] and the flying squirrel (*Glaucomys volans*)[76]. Finally, a proximal centriole was reported in three studied species of family Cricetidae of suborder Myomorpha: golden hamster (*Mesocricetus auratus*)[77], Chinese hamster (*Cricetulus griseus*)[78], and Winkelmann's mouse (*Peromyscus winkelmamzi*)[79]. The presence of a proximal centriole in the three main rodent suborders suggests that a canonical proximal centriole was present in the last common rodent ancestor and is present in many existing rodent species.

In contrast, a proximal centriole is reported to be absent in the five studied members of family Muridae of suborder Myomorpha, which includes the house mouse (*Mus musculus*)[24,56], rat (*Rattus norvegicus*)[43], Mongolian gerbil (*Meriones unguiculatus*), and two *Apodemus* species (*Apodemus agrarius coreae* and *Apodemus speciosus peninsulae*)[80,81]. The location of the missing proximal centriole in the spermatozoan neck of these species is marked by an empty, vault-like space, suggesting that a proximal centriolar remnant may be present. Indeed, a recent study using state-of-the-art cryo-electron microscopy in house mouse spermatozoa confirmed that there were some microtubule remnants where the proximal centriole should be located[56]. The absence of a proximal centriole in multiple genera of family Muridae contrasted with its presence in multiple genera of family Cricetidae suggests that proximal centriole degeneration occurred after the divergence of Muridae from Cricetidae, which happened about 20 million years ago[82].

## Proximal centriole loss correlates with lateral tail attachment to the head in the Muridae ancestor

The observation that a barrel-shaped proximal centriole is present in Cricetidae despite its dispensability post-fertilization suggests that the proximal centriole has a function in the cricetid spermatozoon. Similarly, the observed absence of a barrel-shaped proximal centriole in Muridae, despite its presence in Cricetidae, suggests that the proximal centriole became dispensable in the murid spermatozoon. Since the sperm centriole connects the tail to the head as part of the head-tail connecting apparatus (HTCA)[83] and can coordinate sperm tail and head movement[13], we considered the structure of the proximal centriole in the context of its function in sperm neck and head morphology. Mammalian spermatozoa possess the same basic structural components (head, neck, and tail) but can vary in their morphology, size, and, particularly, in their head shape and the location of the head-neck junction.

Most Eutherian mammals and, specifically, rodents of suborders Hystricomorpha and Sciuromora have a short, oval- or paddle-shaped sperm head with a sperm neck that is attached at or near the center of the head (referred to as a centrally inserted neck)[84] (Fig. 1). As a result, it is thought that the last common rodent ancestor had a paddle-shaped sperm head with a centrally inserted neck[85,86]. Rodent species of suborders Hystricomorpha and Sciuromora also have a proximal centriole, suggesting that these three characteristics (i.e., paddle-shaped head, centrally inserted neck, and the proximal centriole) are associated with each other (Fig. 1c).

While several groups of Myomorpha have the ancestral, paddle-shaped head and centrally inserted neck (i.e., tail attached below the head base center), some have evolved a unique sperm head shape and head-neck attachment point[86,87]. For example, the last ancestor of Cricetidae and Muridae is thought to have evolved a novel, elongated, sickle-shaped head[86] (Fig. 1c). Many species of family Cricetidae have a centrally inserted or off-center head-neck attachment (i.e., tail

attached asymmetrically, below the head base)[85,86] and a proximal centriole, suggesting that their proximal centriole was maintained after the appearance of a sickle-shaped head[77-79]. Therefore, in these cricetids, the presence of a proximal centriole and a centrally inserted neck correlate with each other.

As discussed above, the proximal centriole is absent in family Muridae, and many species in this family exhibit, in addition to a sickle-shaped head, a unique, lateral, head-neck attachment (i.e., tail attached to the side of the head, parallel to the base) (Fig. 1d–e). The difference between the head-neck attachment in Cricetidae and Muridae species suggests that, in rodents, a proximal centriole is present when the neck is centrally inserted (the ancestral form) or off-center and that a proximal centriole is absent when the neck is laterally attached to the head (the derived form), as in Muridae. Therefore, we hypothesize that centrioles are structurally degenerated as part of the evolution of the lateral head-neck attachment in Muridae.

## The primary structure of FAM161A evolved in Muridae and Cricetidae

The spermatozoa of some rodent clades have evolved to successfully fertilize without centrioles. The molecular basis underlying this evolutionary change is unknown, but we hypothesized that it is associated with a change in centrosomal protein structure and function. To identify candidate proteins, we searched for sperm centrosomal proteins that diverge specifically in house mice in contrast to other mammals. We identified sperm centrosomal proteins by comparing the spermatozoan proteome[88–91] to the centrosomal protein database[92]. We performed a rapid and simple protein sequence (i.e., primary structure) comparison between humans and bovine, both of which have two spermatozoan centrioles, and house mice, which lack spermatozoan centrioles; we refer to this calculation as the Identity Ratio (Fig. 2a). A ratio of 1.00 indicates that the human and house mouse orthologs have a percent amino acid identity equal to the ratio between human and bovine; a ratio of more than 1.00 indicates that the human and house mouse orthologs have a greater percent primary structure identity than the ratio between human and bovine; and a ratio of less than 1.00 indicates that the human and house mouse orthologs have a lower percent primary structure identity than the ratio between human and bovine. According to simple phylogenetic considerations (Fig. 2b) and assuming a constant evolutionary rate, we would expect humans to be more identical to house mice than to bovine and, consequently, that their proteins should have an Identity Ratio greater than 1.00.

We found that 450 putative sperm centrosomal proteins had an Identity Ratio range between 0.63 and 1.90, with most centrosomal proteins having an Identity Ratio below 1.00 (a median of 0.99 with a lower 1.5 interquartile of 0.91) (Supplementary Data 2). This low Identity Ratio suggests that mouse centrosomal protein sequences change slightly more than those of bovines and is consistent with previous observations of mutational rate acceleration in the Muridae lineage[93]. Most significantly, FAM161A, a known component of the sperm atypical centriole that exhibits species-specific labeling differences[13], had the third lowest Identity Ratio (0.74) in this list and was far below the lower 1.5 interquartile, suggesting that FAM161A has extensively evolved in house mice (Fig. 2c). To be more stringent in quantifying the level of specific change in house mouse protein sequences, we calculated an Extended Identity Ratio (EIR), which compares the three sequence identity values between house mice and either of three species (humans, bovines, or rabbits) possessing two spermatozoan centrioles to the three sequence identity values between the same species (humans, bovines, and rabbits) (Fig. 2a). Using this quantification, we found an EIR of 0.77 for FAM161A, further supporting its potential role in sperm centriole evolution.

Reproductive genes evolve faster than other genes due to adaptive diversification associated with sperm competition, cryptic female

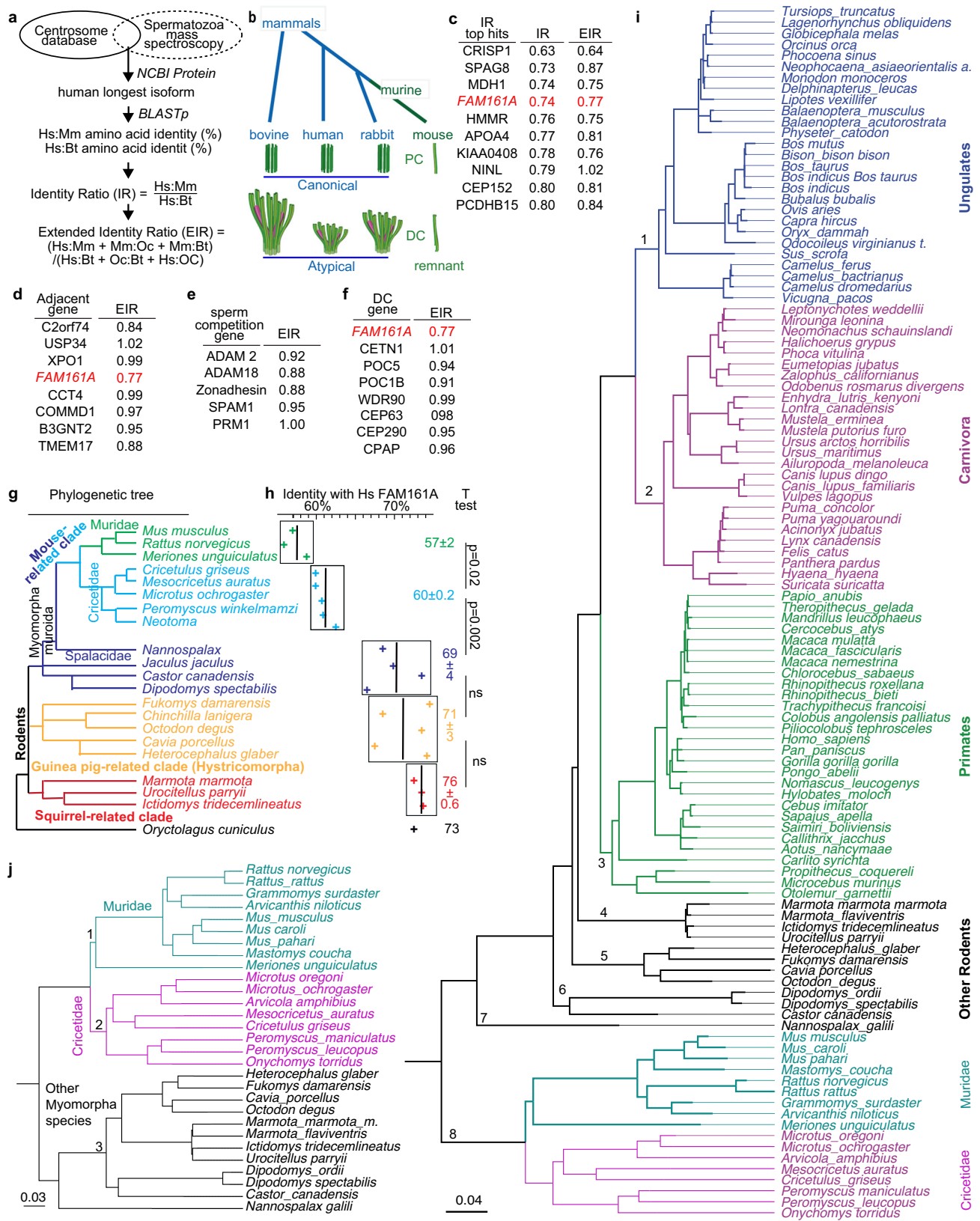

choice, and sexual conflict[94,95]. In contrast, genes integral to sperm function evolve faster in monogamous species than in promiscuous species, possibly because of a reduced selective constraint[96]. Because house mice exhibit a high level of gene sequence diversification[93], the above evolutionary factors may result in a shift in the EIR of FAM161A to below 1.00. Therefore, we tested whether other contributory

evolutionary trends drive the low EIR of FAM161A. In mammals, FAM161A has a paralog, FAM161B, with a higher EIR (EIR = 1.04). Proteins whose genes flank the *Fam161a* gene in house mouse chromosome 11 also had higher EIR (EIR = 0.84–1.02) (Fig. 2d). These values suggest that the *Fam161a* gene evolved without haplotype selection and is an evolutionary hotspot. Finally, sperm proteins known to have

**Fig. 2 | The primary structure of FAM161 is under selective pressure in rodents. a** Calculated extended identity ratios. Hs *Homo sapiens*, Mm *Mus musculus*, Bt *Bos taurus*, Oc *Oryctolagus cuniculus*. **b** Phylogenetic tree showing the evolutionary position of the four mammals used in the extended identity ratio calculations and their proximal (PC) and distal (DC) centriolar structures[82]. **c** The top 10 identity ratio (IR) hits and their extended identity ratios (EIR). Extended identity ratios were calculated for proteins near the FAM161A genomic location (**d**), proteins influenced by sperm competition (**e**), and sperm distal centriolar proteins

(**f**). Rodent phylogenetic tree (**g**) with FAM161A sequence identity relative to human FAM161A (**h**). Percent identity is also shown as average ± SD for individual clades. *$P < 0.05$, **$P < 0.01$ (unpaired, two-tailed *t* test; exact *p*-values are provided in the figure and Source Data File); ns not significant. (**i**) Bayesian phylogeny of FAM161A inferred using nucleotide sequences of mammalian species. (**j**) Bayesian phylogeny of FAM161A inferred using nucleotide sequences of Myomorpha species. The scale bars in **i** and **j** represent the number of nucleotide substitutions per site.

divergent sequences also had higher EIR (EIR = 0.88–1.00) (Fig. 2e)[97]. Altogether, these comparatively higher EIR values suggest that the FAM161A sequence selectively evolved in house mice.

FAM161A belongs to a small set of centriolar inner scaffold proteins that form a unique rod structure in the atypical distal centriole. Yet, the set's other proteins, such as POC1B and POC5, had higher EIR (EIR = 0.91–1.01) than FAM161A (Fig. 2f). Similar results were observed by calculating dN/dS ratio (ω). In this analysis, ω > 1 indicates positive (adaptive, diversifying, increasing amino-acid diversity) selection, ω = 1 indicates neutral evolution, and ω < 1 indicates negative (purifying, the selective removal of deleterious mutations) selection. The median of mammalian protein ω values falls between 0.08 and 0.10, depending on the species being compared[98]. We expected that most centriolar rod proteins would be under purifying selection because centrioles are essential in animal development and physiology[23,99]. Indeed, we found that ω of rodent POC1B and POC5 were close to the mammalian median (ω = 0.19–0.27) (Supplementary Table 1). In contrast, FAM161A had a much higher ω of 0.58, suggesting that the FAM161A sequence is evolving more quickly than other distal centriolar proteins. Consistent with our previous EIR analysis, the ω value of FAM161A was higher in Muridae than in Cricetidae and other Myomorpha species (0.63 > 0.57 > 0.49, respectively), supporting our conclusion that it is rapidly evolving.

To understand the evolution of FAM161A in mammals, we analyzed the FAM161A primary structure identity of humans to other species with a predicted proteome in the National Center for Biotechnology Information (NCBI) protein database (Fig. 2g–h). Human FAM161A has a 70–77% primary structure identity to bovines and rabbits. A similar primary structure identity to human FAM161A was found in species from suborder Sciuromorpha (76 ± 0.6), suborder Hystricomorpha (71 ± 3), and the non-Muridae Cricetidae clades of suborder Myomorpha (69 ± 4). This primary structure identity suggests that basal rodents with centriole-dependent embryonic development, a proximal centriole, and centrally inserted neck have similar FAM161A rates of evolution. In contrast, in the Muridae-Cricetidae clades, FAM161A showed a lower primary structure identity to human FAM161A, at 60 ± 0.2 in Cricetidae species and 57 ± 2 in Muridae species. This statistically significant difference suggests that the FAM161A primary structure started evolving in the Muridae and Cricetidae ancestor (which had off-center and lateral head-neck attachment) and accelerated in the Muridae lineage (species that have centriole-independent embryonic development in common). Considering the other sperm changes (Fig. 1), these findings are consistent with the hypothesis that house mouse FAM161A underwent a distinctive primary structural change that correlates with the emergence of centriole-independent embryo development, lateral head-neck attachment, and proximal centriole loss.

Next, we generated a FAM161A phylogenetic tree of 115 mammals using MrBayes and IQtree (maximum-likelihood and Bayesian phylogenetic trees) (Fig. 2i) and found that the tree had eight main branches. Three large, related branches of non-rodent mammals included primates, carnivora, and ungulates, and exhibited relatively high FAM161A primary structural similarity (branches 1–3 in Fig. 2i). The remaining five branches comprised rodent species (branches 4–8 in Fig. 2i). One basal rodent branch included Muridae and Cricetidae species and was the most diverse group, as expected from accelerated adaptive evolution (branch 8 in Fig. 2i). The remaining four rodent

branches were intermediate between non-rodent mammals and Muridae and Cricetidae species (branches 4–7 in Fig. 2i). Four different methods (Mixed Effects Model of Evolution, MEME; Fixed Effects Likelihood, FEL; Phylogenetic Analysis by Maximum Likelihood, PMAL; and Fast Unconstrained Bayesian AppRoximation, FUBAR) identified many more sites that have undergone adaptive evolution in rodents than in primates, carnivora, and ungulates (Supplementary Table 2). For example, PMAL and FUBAR identified greater than three-fold more positively and negatively selected sites in rodents than in primates, carnivora, and ungulates.

Furthermore, we searched for the rodent subgroup that drives accelerated adaptive evolution in rodents. We generated a FAM161A molecular tree in rodents using FEL (Fig. 2j) and found that the tree has three main branches, which corresponded to Muridae, Cricetidae, and other Myomorpha species. FEL found a similar number of positively selected sites in the three groups (14, 14, and 15 sites, respectively, $P < 0.05$) but a much smaller number of negatively selected sites in Muridae than in Cricetidae and other Myomorpha species (4, 87, and 48, respectively) (Supplementary Table 3).

## Muridae testes express an evolutionarily novel FAM161A protein and mRNA isoform

Mammalian FAM161A has multiple predicted isoforms in the NCBI protein database. Isoforms of two different lengths were studied in humans and house mice[100–102] (Fig. 3a, type 1 and type 2). Here, we refer to the long isoform, comprised of exons 1–7 and expressed in photoreceptors, as type 1 (house mouse 80 KDa, 700 aa, XP_006514891; human 83 KDa, 716 aa, NP_001188472). We refer to the shorter isoform, comprised of exons 1–3 and 5–7 and expressed ubiquitously, as type 2 (house mouse 74 KDa, 644 aa, XP_006514893; human 76 KDa, 660 aa, NP_001188472).

We studied FAM161A in tissue samples by western blot analysis. As expected, house mouse and rat eyes expressed an ~80 KDa protein, corresponding to the FAM161A type 1 isoform (Fig. 3b). Also, expectedly, we observed a single FAM161A isoform of molecular size just above ~72 KDa, probably corresponding to FAM161A type 2, in human and house mouse cells (i.e., U2OS and 3T3) (Fig. 3c). Similarly, in bovine and human testes, we found a single isoform just above ~72 KDa, expected to be FAM161A type 2 (Fig. 3d). These observations confirmed expectations based on published literature[103].

Interestingly, in murids (e.g., house mice and rats), two FAM161A isoforms were expressed in the testes: one at ~72 KDa, which may correspond to type 2, and one shorter isoform below the 72 KDa marker, at ~60 KDa, which we refer to as the type 3 isoform (Fig. 3e). Analysis of house mouse spermatid mRNA shotgun sequencing data predicted that house mouse testes express four transcripts: a most common transcript, NP_001350211, coding for a 555-aa isoform, and an additional three minor transcripts, including XP_006514893, the type 2, 644-aa isoform (Supplementary Fig. 2)[104]. The new, 555-aa isoform has a predicted molecular weight of 64 KDa and is expected to be the type 3 isoform. It contains exons 1, 2, 3, 5, and the beginning of intron 5, which codes for five new amino acids, VVFIGX, followed by a unique, 3′ untranslated sequence (Fig. 3a). We refer to this unique exon 5 as exon 5′. Using 5′ RACE, PCR amplification, and sequencing, we confirmed that the type 3 and type 2 isoforms are expressed in house mouse testes (Supplementary Fig. 2b, c). The type 3 isoform was further

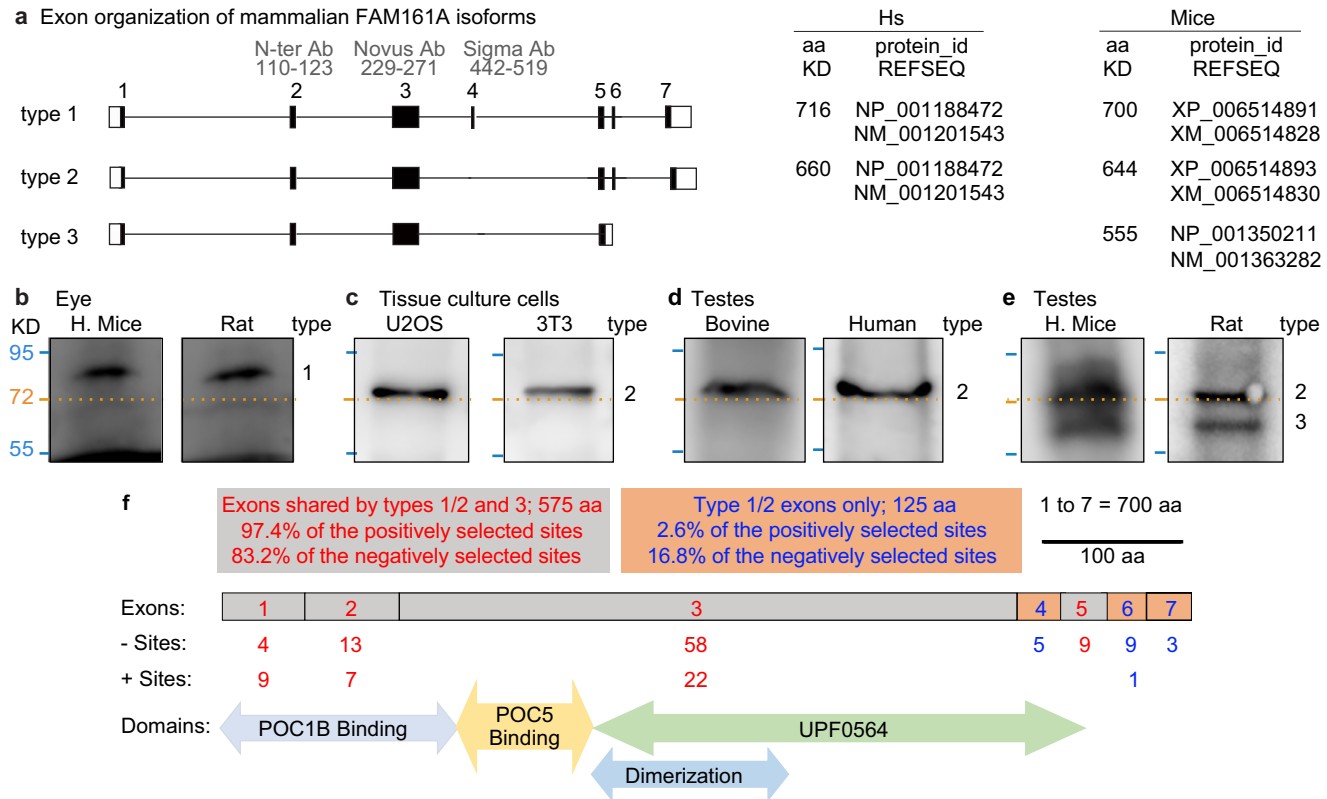

**Fig. 3 | Murids express an evolutionarily novel FAM161A isoform in the testes. a** Exon organization of FAM161A isoform types 1, 2, and 3. ID of predicted FAM161A isoforms and corresponding numbers of amino acids in humans and house mice. **b–f** Western blot analysis of FAM161A in house mouse and rat eyes (**b**), U2OS and 3T3 cells (**c**), bovine and human testes (**d**), and house mouse and rat testes (**e**). Numbers to the right of each blot pair indicate FAM161A isoform types. **f** Diagrammatic representation of *Mus musculus* FAM161A with protein domains, positively (+) selected sites identified by CodeML M8 modeling, and negatively (−) selected sites identified by FEL. All western blot images shown are representative of at least three independent experiments.

confirmed by a search of the NCBI house mouse EST database, which found seven additional FAM161A transcripts with the unique type 3 C-terminus sequence in house mouse testes (Supplementary Fig. 2d).

## The FAM161A type 3 isoform exhibits a relatively high level of positively selected sites

The differential expression of FAM161A isoforms may have resulted in a distinctive rate of evolutionary change. The longest isoform of FAM161A (type 1) has seven exons (700 aa in house mice): exons 1, 2, 3, and 5 (575 aa, or 82.1% of the protein) are found in all isoforms, whereas exons 4, 6, and 7 (125 aa, or 17.9% of the protein) are found only in non-type 3 isoforms. To identify which of these exons are under selection, we employed MEME to identify the sites that are under episodic diversifying selection across the four major mammalian groups (Fig. 3f).

We found that exons 4, 6, and 7 had 17 negatively selected sites out of 125 aa, a rate of 13.6% (Fig. 3f). Similarly, exons 1, 2, 3, and 5 had 84 negative sites out of 575 aa, a rate of 14.6 %. These similar rates suggest a similar level of purifying selection in type 1, 2, and 3 isoforms. In contrast, exons 4, 6, and 7 had only 1 positively selected site out of 125, a rate of 0.8%, whereas exons 1, 2, 3, and 5 had 38 positively selected sites out of 575 aa, a rate of 6.6%. This difference in rates is statistically significant, based on the Z Score Calculator for 2 Population Proportions ($P = 0.01$). The observation that relatively more positive selection occurs in the exons shared between types 1, 2, and 3 than in the exons not expressed in type 3 suggests that the type 3 isoform evolved faster than the other isoforms. Finally, *Mus* and *Rattus* species had few positively (17 and 18, respectively) and negatively (1 and 0, respectively) selected sites as identified by FEL, suggesting that FAM161A changes are fixed and stable in these genera.

## House mouse and human FAM161A isoforms can localize to canonical centrioles

FAM161A is a centriolar protein, and its overexpressed human isoform localizes to the centriole in cultured cells[105]. Therefore, we investigated the localization of overexpressed human type 2, mouse type 2, and mouse type 3 to the centriole in human U2OS and mouse 3T3 cells by labeling the cells with antibodies that recognize centrosomal proteins[106]. Untransfected UTOS cells had one or a pair of pericentrin-labeled foci per cell (21% and 79% of the cells, respectively; $N = 39$). Cells expressing human type 2, mouse type 2, and mouse type 3 FAM161A had FAM161A in most pericentrin-labeled centrioles (Supplementary Fig. 3a, b). Similar results were observed when γ-tubulin marked the centrosome in U2OS cells (Supplementary Fig. 3c, d) or in 3T3 cells labeled against pericentrin (Supplementary Fig. 4a, b) or γ-tubulin (Supplementary Fig. 4c, d). However, in some cases, mouse type 3 had a lower rate of localization to the centrosome (Supplementary Fig. 4d). This finding suggests that, despite the differences in the primary structures of the three isoforms, they are all able to localize to canonical centrioles, although with some differences.

Because FAM161A has different localization patterns in the sperm atypical centriole and canonical centrioles, we also investigated protein interaction outside the canonical centriole by overexpression: see below.

## Human and house mouse FAM161A isoforms have different microtubule and POC5 interactions

Human FAM161A type 1 interacts with human POC5 and POC1B[64], which also interact with each other[64]. Human POC5 also interacts with human CETN protein family members[107]. Of these four proteins, only

FAM161A interacted directly with microtubules (via amino acids 230–543 of type 2, domain UPF0564) (Supplementary Fig. 5a)[105]. Using the FAM161A-microtubule interaction in U2OS cells, we mapped binding between human FAM161A type 2 and POC5 or POC1B through their recruitment to the cellular microtubule network when co-expressed with FAM161A (Supplementary Fig. 5b–f). We complemented our protein interaction mapping with yeast two-hybrid assays and found that amino acids 1–141 of human FAM161A type 2 interacted with POC1B (Supplementary Fig. 5d), amino acids 141–230 of human FAM161A type 2 interacted with human POC5 (Supplementary Fig. 5c–d), and amino acids 1–222 of the human POC5 N-terminus interacted with amino acids 365–478 of the POC1B C-terminus (Supplementary Fig. 5e). These findings suggest a web of interactions between centriolar inner scaffold/rod proteins (Supplementary Fig. 5f).

## House mouse and human FAM161A have distinct subcellular localization patterns to the microtubule cytoskeleton

The interacting domains of FAM161A include most of the positively selected sites in rodents (Fig. 2j). To test for potential functional differences due to divergent primary structure, we expressed house mouse and human FAM161A type 2 in human U2OS cells (Fig. 4). As expected, we found that both proteins localized to the microtubules (Fig. 4a), though they exhibited differences in their subcellular localization patterns and the extent of microtubule co-localization. We have distinguished four types of subcellular localization pattern: "cytoplasmic," with linear FAM161A throughout the cytoplasm; "intranuclear," with FAM161A foci in the nucleus; "perinuclear" (circular), with FAM161A around the nucleus; and "mix intranuclear-perinuclear", characterized by both intranuclear and perinuclear patterns (Fig. 4a). Human FAM161A type 2 mostly localized to the cytoplasmic microtubules. In contrast, house mouse FAM161A type 2 showed a mostly intranuclear or mix intranuclear-perinuclear pattern (Fig. 4a, b). A similar difference in subcellular localization pattern was observed when the two FAM161A type 2 orthologs were expressed in mouse 3T3 cells (Supplementary Fig. 6). These distinct localization patterns suggest that the amino acid differences between human and house mouse FAM161A type 2 produce functional differences.

FAM161A type 3 is 555 amino acids in length and composed of the homologous sequences that mediate binding to microtubules (type 3 amino acids 226–532, domain UPF0564), POC1B (type 3 amino acids 1–139), and POC5 (type 3 amino acids 140–226); however, it is missing the 95 amino acids encoded by type 2 exons 5 and 6, which have unknown function (Supplementary Fig. 7). In contrast to type 2, type 3 mostly formed foci in the nucleus of human U2OS cells (Fig. 4a, b) and mouse 3T3 cells (Supplementary Fig. 6). This localization pattern prompts the hypothesis that the protein domain differences between house mouse FAM161A types 2 and 3 produce functional differences.

## Overexpressed mouse FAM161A type 3 has lower colocalization with overexpressed POC5 outside the centrosomes

RNAi-mediated FAM161A knockdown in U2OS cells had a small effect on centrosomal POC5 localization, suggesting that FAM161A has limited control on POC5 recruitment to the canonical centrioles[108]. Consistent with that, we found that POC5 localized to canonical centrioles when each of the three FAM161A isoforms were overexpressed (human type 2: 100%, $n = 30$; mouse type 2: 100%, $n = 30$; mouse type 3: 100%, $n = 25$) (Supplementary Fig. 8).

Overexpressed POC5 created aggregates in ~39% of transfected cells (Fig. 4d, e). As expected, when human FAM161A type 2 was co-overexpressed with POC5, the POC5 aggregates disappeared in all cells ($N = 30$). Instead, POC5 appeared along the microtubules with FAM161A ("Cytoplasmic," in 60% of transfected cells) and, sometimes, along only some of the microtubules ("Partial cytoplasmic," in 40% of transfected cells) (Fig. 4d, e). However, when mouse FAM161A type 2

was co-overexpressed with POC5, the POC5 colocalized with FAM161A in the aggregates and cytoplasm (in 40% of transfected cells) as well as around the nucleus (in 60% of transfected cells). Interestingly, when mouse type 3 was co-overexpressed with POC5, the POC5 colocalized with FAM161A in the centrosomes, but only 50% of POC5 aggregates contained FAM161A (N aggregates = 497, N cells = 12). We quantified overexpressed POC5 colocalization with various FAM161A isoforms and found that POC5 colocalized significantly less with mouse type 3 FAM161A than with overexpressed human and mouse type 2 (Fig. 4f). This reduced colocalization suggests that mouse type 3 has a weaker interaction with POC5 than mouse type 2.

## Overexpressed mouse FAM161A type 3 can prevent overexpressed FAM161A type 2 and POC1B from localizing to the cytoplasmic microtubules

Human FAM161A type 2 forms homodimers via amino acids 230–386[105]. This amino acid sequence corresponds to amino acids 226–380 of house mouse types 2 and 3, suggesting that they can also dimerize. To gain insight into the effects of house mouse types 2 and 3 on each other, we co-expressed them in U2OS cells. We found that type 3 prevented type 2 from colocalizing with the circular microtubules (perinuclear pattern) and, instead, targeted type 2 to foci in the nucleus (intranuclear pattern) (Fig. 4a–c). This observation suggests that mouse FAM161A type 3 can act as a dominant negative isoform that prevents FAM161A type 2 from localizing to the cytoplasmic microtubules.

Human FAM161A type 2 interacts with POC1B and recruits it to the cytoplasmic microtubules[109]. Therefore, we next examined the interaction of house mouse FAM161A with mCherry-POC1B by coexpressing them in U2OS cells. Usually, rod proteins (i.e., POC1B, POC5) form aggregates when expressed in U2OS cells (without expressing FAM161A), because these proteins interact with each other (Supplementary Fig. 5a). As expected, we found that house mouse FAM161A type 2 recruited POC1B to the microtubules (Supplementary Fig. 9a). However, FAM161A type 3 did not recruit POC1B to the microtubules but instead colocalized with POC1B aggregates (Supplementary Fig. 9b–e). This suggests that both isoforms maintained the ability to interact with POC1B, which is expected, since the FAM161A N-terminal domain that mediates the interaction with POC1B is found in both isoforms (Supplementary Fig. 5f). Finally, co-expressing type 2 with type 3 inhibited type 2 from recruiting POC1B to the microtubules (Supplementary Fig. 9c). These observations suggest that FAM161A type 3 can interact with both FAM161A type 2 and POC1B but specifically lost the ability to interact with microtubules and may act as a dominant negative in vivo. Altogether, the above in vitro observations suggest that house mouse testes express FAM161A proteins that localize to canonical centrioles but are functionally distinct from human FAM161A.

## Cricetidae and Muridae spermatozoan FAM161A is present only in the distal centriole

FAM161A is a component of the distal and proximal centrioles in basal mammals, such as humans, bovines, rabbits, and dogs[13] (Supplementary Fig. 10). To gain insight into the localization of FAM161A in rodents, we stained mature spermatozoa isolated from the caudal epididymis of house mice, *Peromyscus maniculatus* (deer mouse; family Cricetidae, subgroup Neotominae), and *Microtus ochrogaster* (prairie vole; family Cricetidae, subgroup Arvicolinae) using FAM161A Ab1, POC1B Ab2, and tubulin antibodies (Fig. 5a–c).

In the house mouse spermatozoan neck, FAM161A labeled a single focus near the axoneme tip, which most likely represents a distal centriolar remnant (Fig. 5a). The FAM161A focus was uniquely short and wide, unlike in humans, bovines, and rabbits, where it is longer, indicating a distinct localized distribution. FAM161A and tubulin labeling were undetected in the proximal centriole (Fig. 5a). POC1B

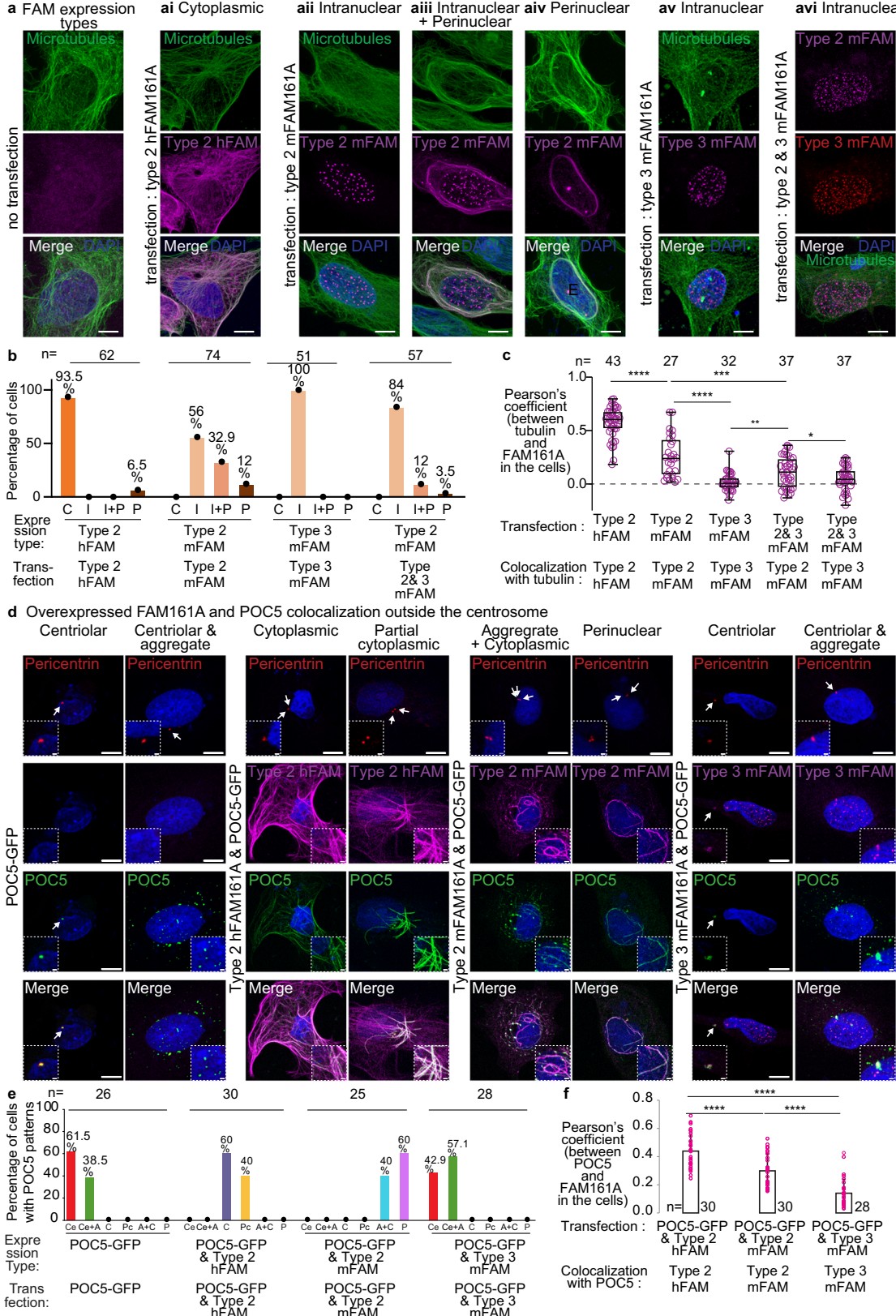

labeled a single focus proximal to the nucleus that most likely corresponds to the proximal centriolar remnant. This data suggests that despite centriolar ultrastructural loss in house mice, some centriolar inner scaffold proteins are still present, though in a unique pattern.

Like in house mice, in deer mice, FAM161A labeled a single, short, and wide focus on the distal centriole, indicating that house mice and deer mice have similar FAM161A localization patterns (Fig. 5b). Also, like in house mice, tubulin was not detected in the neck of deer mouse sperm. However, in contrast to house mice, POC1B in deer mice distinctly labeled two foci, as observed in most mammals, which we believe represent the proximal and distal centrioles. Overall, FAM161A and POC1B localization in deer mice show

**Fig. 4 | Human and house mouse FAM161A isoforms have different microtubule and POC5 interactions. a** Expression analysis of human FAM161A (hFAM161A) type 2 (second panel from left), house mouse FAM161A (mFAM161A) type 2 (middle three panels), mFAM161A type 3 (second panel from right), and the combination of mFAM161A types 2 and 3 (right panel) in U2OS cells. **b** Quantification showing the percentage of cells exhibiting each of the various expression patterns observed during expression analysis of hFAM161A type 2, mFAM161A type 2, mFAM161A type 3, and the combination of mFAM161A types 2 and 3. C, "Cytoplasmic"; I, "Intranuclear"; I + P, "Mix intranuclear-perinuclear"; P, "Perinuclear". **c** Quantification of FAM161A colocalization with tubulin. ****$P < 0.0001$, ***$P < 0.001$, **$P < 0.01$, *$P < 0.05$ (unpaired, two-tailed $t$ test; exact p-values are provided in the Source Data File); ns not significant, n number of cells, scale bars are 8 μm. The data shown are the representative images and compiled quantification from three independent

experiments. Data are presented as box and whisker plots, where upper and lower bounds show interquartile range, the line within the box shows the median, and whiskers show minimum and maximum data points. **d** Co-overexpression of FAM161A isoforms and POC5 in U2OS cells. The inset in the bottom left corner shows a zoomed view of the site of the centriole, and the inset in the bottom right corner shows a zoomed view of non-centriolar POC5 locations in the cell. Scale bars are 8 μm, inset scale bar is 1 μm. **e** Quantification showing the percentage of cells exhibiting the various expression patterns observed during expression analysis of hFAM161A type 2, mFAM161A type 2, and mFAM161A type 3. Ce, "Centriolar"; Ce+A, "Centriolar + Aggregate"; C, "Cytoplasmic"; Pc, "Partial Cytoplasmic"; A + C, "Aggregate + Cytoplasmic"; P, "Perinuclear". **f** Quantification of FAM161A colocalization with POC5. Statistical analysis used was unpaired, two-tailed $t$ test. Source data are provided in the Source Data File.

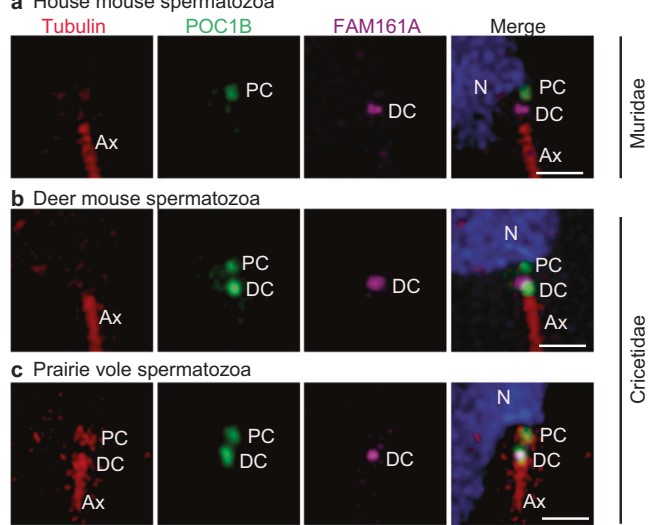

**Fig. 5 | Cricetidae and Muridae species share an evolutionarily novel FAM161A localization in their spermatozoa.** Rod protein localization in mature spermatozoan necks of house mice (**a**), deer mice (**b**), and prairie voles (**c**). Tubulin is used as a marker for the centriole and axoneme. PC proximal centriole, DC distal centriole, Ax axoneme. Scale bars are 1 μm. The images shown are representative of three independent experiments.

characteristics that are intermediate between most mammals and house mice.

Finally, in prairie voles, which belong to a different Cricetidae subgroup, tubulin was detected in the two spermatozoan centrioles, and FAM161A labeling was similar to that of deer mice and house mice (i.e., showing a wide focus on the distal centriole) (Fig. 5c). Overall, FAM161A shows an evolutionarily novel, single-focus labeling pattern in Muridae and Cricetidae, suggesting that a change in FAM161A localization took place in the common ancestor of these families. Also, these findings support the hypothesis that sperm centrioles gradually evolved in the superfamily Muroidea and that proximal centriole degeneration is most extreme in the family Muridae.

### Humans, rabbits, and bovines share a conserved centriole remodeling program

Mammalian spermatozoan centrioles are remodeled from canonical proximal and distal centrioles during spermiogenesis in the seminiferous tubule of the testes[24,63]. To study FAM161A changes during mammalian evolution, we first studied its localization relative to three other typical centriolar inner scaffold and atypical distal centriolar rod proteins (CETN1, POC5, and POC1B) in basal mammals (with two spermatozoan centrioles), namely humans, rabbits, and bovines. To do this, we performed immunofluorescence (Fig. 6a–c) and quantified the

total centriolar relative labeling intensities (Fig. 6d–f). We looked at three sperm cell stages: pre-haploid sperm (spermatogonia and spermatocytes), round spermatids, and elongated spermatids. These stages were identified based on the location of cells relative to basal lamina in the seminiferous tubules, staining with lectin PNA, and the shape of the nucleus (Fig. 6ai, bi, ci).

In pre-haploid sperm cells, CETN1, POC5, POC1B, and FAM161A colocalized in the two centrioles (**Sg in** Fig. 6a–c). POC1B was detected only with POC1B antibody 2 (Ab2) in pre-haploid sperm of bovine and rabbits (**Sg in** Fig 6aiii–iv & Fig 6biii–iv), though this was likely due to a lower reactivity of POC1B antibody 1 (Ab1) in these species and the generally lower immunoreactivity of POC1B at this cell stage (**Sg in** Fig. 6d–e). Like POC1B, CETN1, POC5, and FAM161A also showed the lowest immunolabeling intensity at this sperm cell stage in all species, suggesting that later stages are characterized by enrichment of all four proteins (**Sg in** Fig. 6d–f). Additionally, the length of CETN1 labeling across all three species was similar (~310 nm), suggesting that before centriole remodeling, the centriolar inner scaffold has an evolutionarily conserved size (Supplementary Fig. 11).

In round spermatids, CETN1, POC5, POC1B, and FAM161A labeling increased in length in the two centrioles (**RS in** Fig. 6a–c). Accordingly, the labeling intensity of all four proteins was higher at this stage than at the earlier pre-haploid stage in all three species (**RS in** Fig. 6d–f). In rabbits, FAM161A labeling was undetectable, which was likely due to a weaker immunoreactivity of the FAM161A antibody in this organism. In all three species, CETN1 labeling in the round spermatid proximal and distal centrioles was longer than in pre-haploid sperm centrioles, and distal centriole labeling was longer than that of the proximal centriole (Supplementary Fig. 11). The length of CETN1 in the round spermatid proximal and distal centrioles was species-specific, being longest in bovine ($451 \pm 62$ and $660 \pm 72$ nm, respectively), intermediate in rabbit ($409 \pm 40$ and $477 \pm 72$ nm, respectively), and shortest in humans ($356 \pm 59$ and $378 \pm 46$ nm, respectively) (Supplementary Fig. 11). Together, these findings in round spermatids suggest that the first stage of remodeling and enrichment of the four rod proteins in the proximal and distal centrioles is conserved, but the size extension is species-specific.

In elongated spermatids, CETN1 and POC5 immunostaining intensity levels in the two centrioles were maintained or declined relative to their levels in round spermatids in all three species (**ES in** Fig. 6d–f). In contrast, POC1B and FAM161A showed more intense labeling in elongated spermatids than in round spermatids of all three species (**ES in** Fig. 6d–f), while CETN1 showed an inconsistent staining pattern (**ES in** Fig. 6a–c). Proximal centriole labeling by the four proteins (which was extended in round spermatids) was shortened in elongated spermatids. In the elongated spermatid distal centriole, POC5 reorganized into a "V"-shape in bovine and a wide focus in rabbits and humans (**ES in** Fig. 6a–c). POC1B labeling took on a filled "V"-shape in the distal centriole of bovine elongated spermatids and a wide focus in the distal centriole of rabbit and human elongated spermatids (**ES in** Fig. 6a–c). FAM161A labeling reorganized into a "V"- shape in the

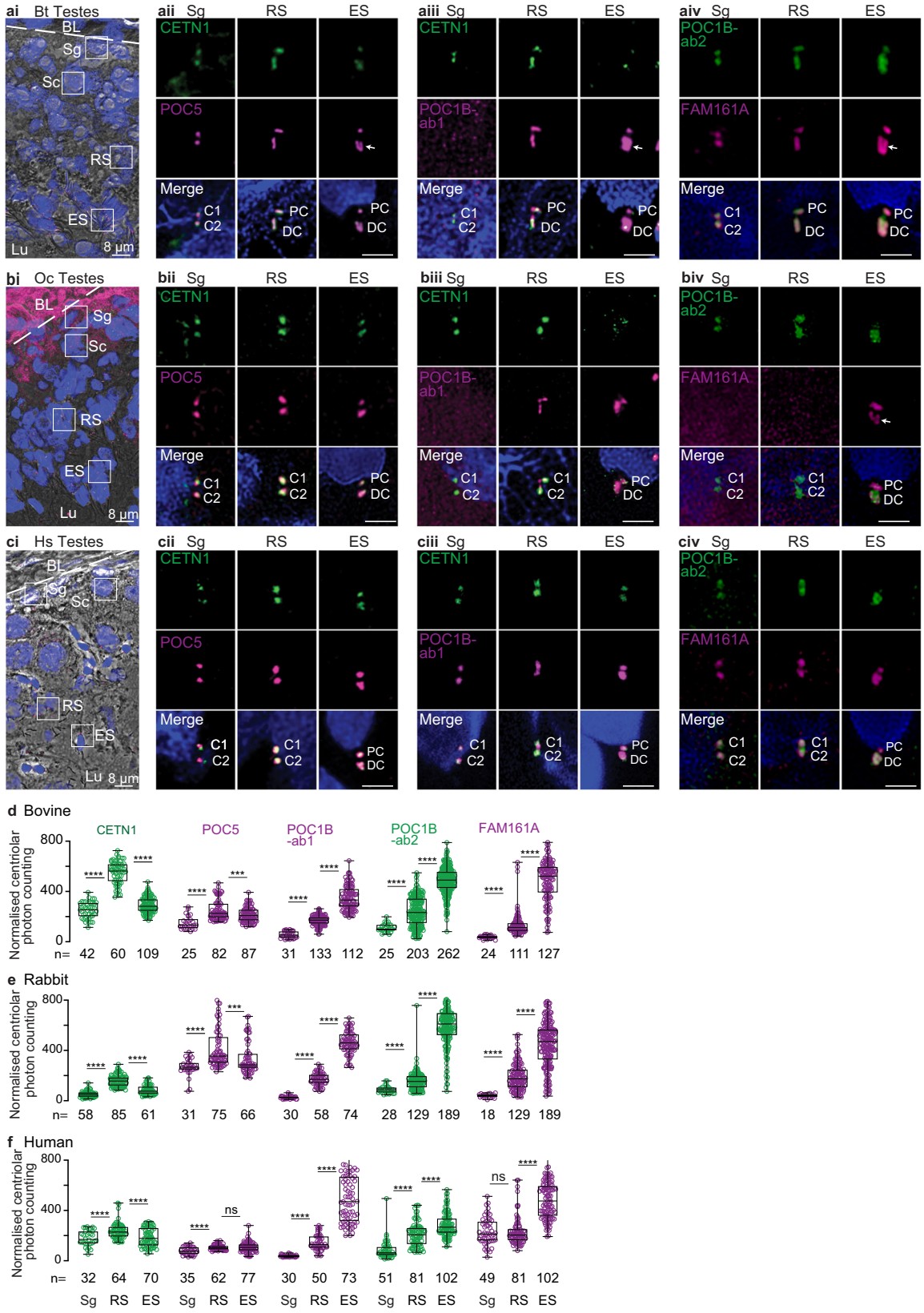

distal centriole of bovines and rabbits and into a wide focus in humans (**ES in** Fig. 6a–c). Together, these findings suggest that the second stage of remodeling is also conserved and includes shortening toward their original length of the four rod proteins in the proximal centriole and the formation of a wider distribution and appearance of two rods in the distal centriole.

Quantification of CETN1, POC5, POC1B, and FAM161A in individual proximal and distal centrioles during bovine spermatogenesis found localization intensity changes that are similar to what is observed during total centriole quantification (Supplementary Fig. 12).

Overall, the spermatogenesis findings demonstrate that rod proteins share two conserved remodeling steps with some species-specific

**Fig. 6 | Bovines, rabbits, and humans share a conserved centriole remodeling program. a–a** A single seminiferous tubule section showing various stages of spermatogenesis in bovines (**ai**), rabbits (**bi**), and humans (**ci**). The white dotted line indicates the basal lamina boundary. *Throughout the paper*: BL basal lamina, Sg spermatogonia, Sc spermatocyte, RS round spermatid; Es elongated spermatid, Lu lumen. Scale bars are 8 μm. Representative images of various rod proteins at various stages of spermatogenesis in bovines (**aii–iv**), rabbits (**bii–iv**), and humans (**cii–iv**). Scale bars are 2 μm. Quantification of total centriolar localization, including various proximal and distal centriolar proteins at various stages of sperm development in bovines (**d**), rabbits (**e**), and humans (**f**). The data was generated from three independent experiments. C1/2, centrioles 1 and 2; PC proximal centriole, DC distal centriole, Bt *Bos Taurus* (bovine), Oc *Oryctolagus cuniculus* (rabbit), Hs *Homo sapiens* (human), n, sample size. The two centrioles in Sg are labeled as C1 and C2 since the proximal and distal centrioles are not phenotypically distinguishable at this stage. The white arrow marks the "V"-shaped rods or filled-in "V" shape. The graphs are presented as box and whisker plots, where upper and lower bounds show interquartile range, the line within the box shows the median, and whiskers show minimum and maximum data points. \*\*\*\*$P < 0.0001$, \*\*\*$P < 0.001$, \*\*$P < 0.01$, \*$P < 0.05$ (unpaired, two-tailed $t$ test; exact p-values are provided in the Source Data File); ns not significant, n number of cells. Data shown are the representative images and compiled quantification from at least three independent experiments. Source data are provided in the Source Data File.

length differences. This conservation in distantly related mammalian groups suggests that this program is ancestral in mammals. As part of this program, FAM161A is gradually recruited to the proximal and distal centrioles during spermiogenesis.

### The bovine distal centriole begins splaying in round spermatids and is asymmetric in elongated spermatids

This remodeling program may have additional characteristics that are difficult to appreciate with confocal microscopy alone. Therefore, we next performed STORM microscopy in bovine to better understand the organization of centriole remodeling.

Among the species we studied, bovines had the longest round spermatid centrioles. Therefore, we used high resolution N-STORM microscopy to collect more detailed information about their proximal and distal centrioles (Supplementary Fig. 13a). We confirmed that the round spermatid proximal centriole is longer than the pre-haploid proximal centriole and that the round spermatid distal centriole is longer than the round spermatid proximal centriole (Supplementary Fig. 13b). We noticed that the rostral end of the round spermatid distal centriole was 20–23% wider than its caudal tip, as labeled by CETN1, POC5, and POC1B (Supplementary Fig. 13c), suggesting that the distal centriole begins splaying in parallel to rod protein extension in round spermatids. Note that at the round spermatid stage, the proximal centriole elongates to form the centriolar adjunct, which is a short axoneme-like structure that appears during spermiogenesis[110,111]; this indicates that the rod proteins may extend into the centriolar adjunct. When elongated spermatids are oriented such that the proximal centriole is on the right side, we observed, using STORM, that the left rod of the bovine distal centriole is longer and wider than the right rod, indicating that distal centriole asymmetry arises as part of centriole remodeling during spermiogenesis (Supplementary Fig. 13d). Altogether, these findings suggest that the distal centriole begins splaying in round spermatids and that rod asymmetry is apparent in elongated spermatids.

### Deer mice followed the ancestral centriole remodeling program, but house mice pivoted to a centriole degradation program

The primary structure and spermatozoan localization of FAM161A suggest that the family Cricetidae has intermediately modified sperm centrioles. Therefore, we next analyzed centriolar inner scaffold protein localization during spermatogenesis in deer mice (*Peromyscus maniculatus*), a Cricetidae species with a recognizable proximal centriole and modified FAM161A. We found that CETN1, POC5, POC1B, and FAM161A labeling in deer mouse spermatids was consistent with the ancestral remodeling program observed in the basal mammals (i.e., bovines, rabbits, and humans) (Fig. 7a, b). This finding supports the hypothesis that Cricetidae species undergo a remodeling program that is similar to most other mammals.

Next, we analyzed centriolar inner scaffold protein localization during spermatogenesis in house mice (Fig. 7c–e and Supplementary Fig. 14). House mouse spermatozoa contain two highly modified centrioles (remnant centrioles) that are characterized by the lack of CETN[18,24], a proximal centriole comprised of a few doublet or triplet microtubules, and a distal centriole comprised of the central pair[56].

In house mouse pre-haploid sperm cells, like in bovines, rabbits, humans, and deer mice, CETN1 and POC5 co-localized in the two centrioles (**Sg in** Fig. 7cii). POC1B and FAM161A were undetected in house mouse pre-haploid sperm cells (**Sg in Fig 7ciii–iv)**, which was likely due to the antibodies having a weaker immunoreactivity in house mice than in other species and to the levels of these proteins being lower at this sperm cell stage than at other stages (**Sg in** Fig. 7cii–iv and Fig. 7d).

In house mouse round spermatids, like in bovines, rabbits, and humans, CETN1, POC5, and POC1B labeled the proximal and distal centrioles (**RS in** Fig. 7cii–iii), and the length of the labeling was longer in the distal centriole than in the proximal centriole and pre-haploid centrioles (Fig. 7e). Also, like in bovines, rabbits, and humans, the labeling intensity of the three proteins was higher in round spermatid centrioles than in earlier, pre-haploid sperm cells (Fig. 7d). Unlike in bovines, rabbits, humans, and deer mice, house mouse FAM161A was present as one or two dot-like structures near the proximal centriole-distal centriole junction (**RS in** Fig. 7civ), and a similar labeling pattern was observed using a second FAM161A antibody (Supplementary Fig. 14). Together, these round spermatid findings suggest that most aspects of the first stage of centriole remodeling in house mice (centriolar elongation and enrichment of CETN1, POC5, and POC1B in the distal and proximal centrioles) are similar to those observed in other mammals.

In house mouse elongated spermatids, rod protein localization was different between early and late elongated spermatids; this difference was not observed in bovines, rabbits, and humans. Like in bovines, rabbits, nor humans, CETN1 and POC5 were present in house mouse early elongated spermatids (**eES in** Fig. 7cii–iii); however, unlike in bovines, rabbits, and humans, CETN1 and POC5 were undetectable in the centrioles of house mouse late elongated spermatids. (**lES in** Fig. 7cii–iii). Also, distinct from bovines, rabbits, humans, and deer mice, POC1B and FAM161A were not enriched in elongated spermatids relative to round spermatids. Rather, FAM161A was observed as a pair of dots between the POC1B-labeled proximal and distal centrioles in late elongated spermatids (**eES & lES in** Fig. 7civ). Together, these findings in house mouse elongated spermatids suggest that the second centriole remodeling stage (centriolar enrichment of POC1B and FAM161A) is dramatically modified in house mice and consistent with a reduction in rod proteins.

Overall, our findings during spermatogenesis reveal that, in general, centriolar inner scaffold proteins in house mouse spermatids begin the remodeling process like in other mammalian species but then pivot to becoming reduced in elongated spermatids. This pivot suggests that the house mouse undergoes a derived remodeling program called centrosome reduction (i.e., degradation)[112], which appears to lead to the absence of sperm centrioles in this organism. In this centrosome reduction (i.e., degradation) program, FAM161A labels only one structure found at the proximal centriole-distal centriole junction.

Quantification of CETN1 and POC5 in individual proximal and distal centrioles during house mouse spermatogenesis found localization intensity changes that are similar to what is observed during total centriole quantification (Supplementary Fig. 15).

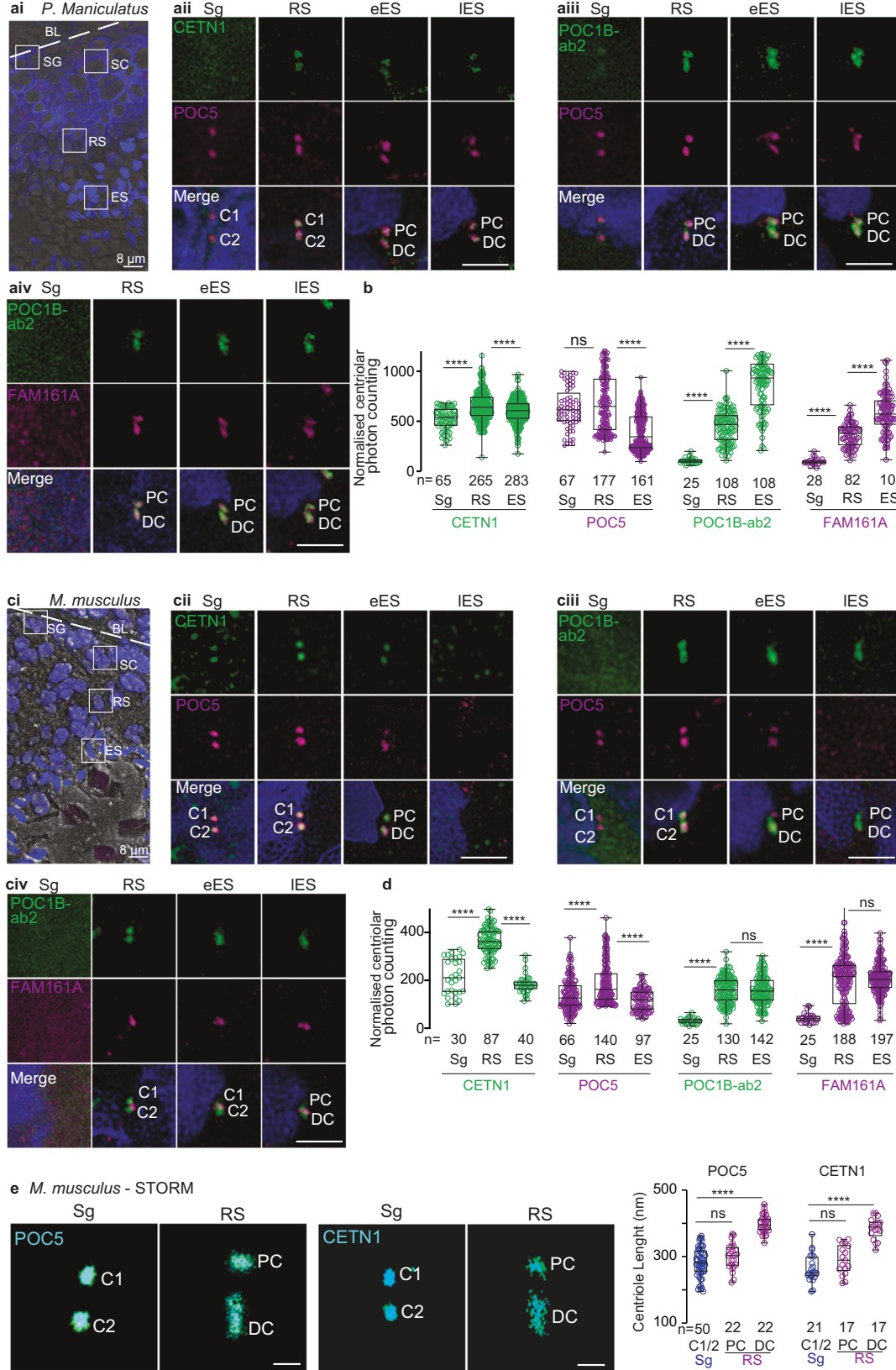

## Discussion

Sperm morphology is continuously and rapidly evolving, particularly in internally fertilizing species, as part of a large-scale evolutionary sex cascade[7,29,36,113]. This morphological evolution includes the sperm head, tail, and neck, a strategic location connecting the most significant functional units of the spermatozoon[114–116]. The simple organization of sperm neck structures is conserved in the spermatozoa of external fertilizers[117] but has evolved diverse forms in internally fertilizing species, presumably as part of a reproductive centriole evolutionary cascade[114,118]. In mammals, the sperm neck contains specialized and unique centriolar structures, such as the atypical distal centriole and striated columns in basal mammals, which

**Fig. 7 | Unlike the deer mouse remodeling program, the house mouse centriole degradation program begins in elongated spermatids. a, c** A single seminiferous tubule section showing various stages of spermatogenesis in deer mice (**ai**) and house mice (**ci**). The white dotted line indicates the basal lamina boundary. BL basal lamina, Sg spermatogonia, Sc spermatocyte, RS round spermatid, Es elongated spermatid, Lu lumen. Scale bars are 8 μm. Representative images of various rod proteins at various stages of spermatogenesis in deer mice (**aii–iv**) and house mice (**cii–iv**). **b, d** Quantification of the combined proximal and distal centriolar localization of various proteins at various stages of sperm development in deer mice (**b**) and house mice (**d**). **e** STORM imaging of house mouse testes with POC5 and

CETN1 staining in spermatogonia/spermatocytes (Sg) and round spermatids (RS) (left panels). Measurements of centriolar length as determined using STORM imaging (right panel). Scale bars are 0.4 μm. The data were generated from three independent experiments; n number of cells in **b** and **d**, and number of centrioles in **e**. The graphs are presented as box and whisker plots, where upper and lower bounds show interquartile range, the line within the box shows the median, and the whiskers show minimum and maximum data points. ****$P < 0.0001$, ***$P < 0.001$, **$P < 0.01$, *$P < 0.05$ (unpaired, two-tailed $t$-test; exact $p$-values are provided in the Source Data File). Source data are provided in the Source Data File.

include humans, and remnant centrioles in murids[15,56,63]. Here, we hypothesize that in murids, centriolar structure and the centriolar protein FAM161A evolved rapidly in correlation with centriole dispensability for early embryogenesis and acceleratingly with the change in neck attachment to the side of the sperm head. While it was known that atypical centrioles evolved through the remodeling of canonical spermatid centrioles during spermatogenesis after they form the sperm flagella, here, we find that degenerated centrioles in these rodents evolved through modification of the centriole remodeling program.

We found that a two-step centriole remodeling program that forms the atypical centriole in humans, rabbits, and bovines is modified, leading to centriole degeneration in murids. We also found that the evolutionary transition from centriole remodeling to centriole degeneration was a gradual process of rodent sperm evolution, during which centriolar composition changed before distal centriolar structure became unrecognizable in murids. We found that some non-murid rodents, such as deer mice and voles, have centrioles that are intermediate between those of humans and house mice, suggesting that they are a signature of an evolutionary transition. Additionally, we found that the alteration in the centriole remodeling program that leads to centriole degeneration in house mice evolved in correlation with a change in the sequence and function of FAM161A. We propose that centriole remodeling began to evolve in the rodent suborder Myomorpha, in the ancestor of Muridae and Cricetidae, at or near the time of Eumuroida (note that Eumuroida is a new classification that includes additional families about which we have no information regarding their sperm centrioles[119]). Later, though prior to Muridae, centriole remodeling gradually transitioned to centriole degeneration through modification of the ancestral mammalian centriole remodeling program.

Successful reproduction depends on the competitiveness and efficiency of the sperm cell and the fecundity of the female. The rapid evolution of sperm morphology and ultrastructure is often driven by sperm competition[113]. However, adaptations that yield competitive fertilization success may come at a cost to the female or the resultant embryo[120,121]. A recent study showed that in humans and bovines, the centrioles are associated with aneuploidy in the developing embryo and early miscarriages, suggesting that centriole remodeling may reduce fecundity[41] and that atypical centrioles are associated with reduced reproductive success[122]. We propose that the evolutionary transition of centrioles with a canonical structure in basal vertebrates and Tetrapods, to an atypical structure in the spermatozoa of most mammals, to a remnant ultrastructure in house mice may have been driven by sperm competition in the rodent lineage. These changes describe a sperm centriole evolutionary cascade that transformed the sperm centriole from an essential structure to a dispensable one. We predict that other r-strategy mammals would have evolved mechanisms to overcome the costs associated with the atypical centriole. Considering that Cricetidae and Muridae species undergo centriole-independent embryo development, it is tempting to speculate that the dispensability of the sperm proximal centriole for post-fertilization function relieved it from some evolutionary constraints, allowing it and the head-neck attachment to change.

The evolutionarily conserved functions of the centriole are to form a cilium and mediate several structural functions in the ciliary life cycle[123]. The centriole forms the cilium by first nucleating the axoneme, with the nine-fold symmetry of the centrioles providing the template for the nine doublets of the axoneme. Then, after the cilium forms, the centriole anchors it to the cell, regulating its motile function and, together with the ciliary transition zone, serving as an entry site to the compartmentalized cilium. Finally, after the cilium disassembles during cell division, the centriole and its linked daughter centriole are incorporated into a centrosome to ensure accurate inheritance by the two daughter cells. These critical functions impose limits on centriolar structural evolution. Changes in centriolar structure that occur before the axoneme form result in an axoneme with a non-canonical design. Indeed, the giant sperm centriole of *Sciara* is composed of a uniquely large number of microtubules, producing unique axoneme structures[124]. In contrast, changes to centriolar structure that do not result in changes to the axoneme can occur after the axoneme is assembled, a process suggested to be a form of "organelle heterochrony"[125]. Such a change (i.e., centriole remodeling) occurs in most mammals with an atypical distal centriole, which acts as a dynamic basal complex[13]. As described above, a more dramatic change to centriolar structure occurs in Muridae, in which both the centriole at the flagellum base and its associated centriole degenerate after the axoneme is formed[24]. Here, we suggest that this degeneration evolves as an extension of centriole remodeling and as an adaptation to changes in sperm-neck attachment.

Sperm centrioles play several critical functions in the early embryo after fertilization[126], including forming the sperm aster that brings the male and female nuclei together[127] and serving as a nucleation site for new centrioles[17]. These functions are achieved through alternate mechanisms in the mouse embryo, which does not inherit its centrioles from the sperm. Maternal microtubule organizing centers, together with the actin cytoskeleton, bring the male and female pronuclei together[47,49–51,128], and embryonic centrioles appear de novo at the blastula stage[22,47]. This appearance of the centrioles at the blastula stage is needed so that cells can generate cilia, which are essential during development[48]. In the future, it will be important to study the evolution of these mechanisms in the ancestor of Muridae and Cricetidae.

We also propose that the degradation of the centrioles in house mice is partly attributable to the position of attachment between the sperm head and tail. This change in position may have coevolved with the female reproductive tract[40,129–131]. Species with paddle-shaped sperm heads and tail attachment to the base of the head (most mammals, including humans, rabbits, and bovines) have atypical centrioles. Most of the rodent species with hooked sperm heads and an off-center tail attachment to the base of the head have centrioles with intermediate composition conservation. In contrast, murids, which have hook-shaped sperm heads and tail attachment on one side of the head, have remnant centrioles. This innovation in murid sperm head and neck morphology may have been fueled by the dispensability of the centrioles.

Here, we show evidence supporting three main conclusions: (i) the loss of sperm neck centrioles is a derived feature of Muridae; (ii)

FAM161A and, in particular, the type 3 isoform are interesting candidates for functional tests concerning centriole degradation; and (iii) house mouse FAM161A type 2 and type 3 isoforms function differently from the human type 2 isoform, in a manner consistent with their important role in centriole degradation. Further exploration of these conclusions can be accomplished by combining site-directed mutagenesis and comparative biology.

For example, Cricetidae and Muridae spermatozoan FAM161A is present only in the distal centriolar remnant. In contrast, Cricetidae POC1B is present in both centrioles, while Muridae spermatozoan POC1B is present mainly in the proximal centriolar remnant. This indicates a gradual progression from ancestral to degenerate centrioles in the evolution of Muridae from an ancestor shared with Cricetidae. It would be interesting to track this gradual evolution by examining the localization of these two proteins in various Muridae genera. Additionally, to test the role of the centriolar structure in the sperm neck attachment position, it would be interesting to study whether centriolar structure and protein changes occurred independently in other lineages (convergent evolution) and in murid species that revert to a central neck attachment position.

We found that mouse and human FAM161A show distinct subcellular localization patterns to the microtubule cytoskeleton. It would be interesting to define the molecular basis of this difference by using site-directed mutagenesis to humanize the distinct domains of the mouse FAM161A amino acid sequence.

We also found that overexpressed mouse FAM161A type 3 can prevent overexpressed FAM161A type 2 and POC1B from localizing to the cytoplasmic microtubules. It would be interesting to define the molecular basis of this dominant negative behavior and to determine whether this behavior is due to the unique, five-amino-acid (VVFIGX) motif at the end of exon 5 or to the lack of exons 4, 6, and 7.

Finally, we found that mouse FAM161A differs from human FAM161A in two ways, which might explain sperm centriole degradation in Muridae: (1) primary structural differences between type 1 and type 2 isoforms; and (2) expression of the type 3 isoform in the testes. To determine the relative contribution of these differences to sperm centriole degradation, we propose, first, replacing mouse FAM161A with human FAM161A and, second, fusing exon 5 and exon 6 to specifically prevent expression of the type 3 isoform.

Collectively, our findings provide molecular evidence of rapid diversification and adaptive evolution of centrioles and the neck region in mammalian sperm. In the future, it will be important to identify the specific selective pressures that brought about mammalian centriole remodeling and centriole dispensability in some rodents.

## Methods

### Ethical statement
The authors are accountable for all aspects of the work and for ensuring that questions related to the accuracy or integrity of any part of the work are appropriately investigated and resolved.

The institutional review boards of the University of Toledo, Cornell University, and University of Maryland approved all experiments involving animal samples. All experiments were performed in accordance with relevant guidelines and regulations.

The University of Toledo institutional review board (IRB #0000202366) approved all experiments involving human samples, and informed consent was obtained from all human research participants.

Researchers from multiple US states and two continents were included throughout the research process, including study design, undertaking, data analysis, and authorship. All experiments were carried out in strict compliance with the relevant laws and with the approval of the Scientific Ethics Committees of the respective universities. All human research participants have signed a consent form. All roles and responsibilities were agreed upon by collaborators.

### U2OS cells
U2OS cells were purchased from American Type Culture Collection (cat # CRL-3455). Cells were maintained in Dulbecco's Modified Eagle's Medium (DMEM; Mediatech) supplemented with 10% fetal calf serum (FCS; Atlanta Biologicals) and kept at 37 °C in a humidified atmosphere supplemented with 5% $CO_2$.

### Caudal sperm isolation
Caudal sperm were isolated from sexually mature captive-bred house mice (*Mus musculus* used in a research lab setting), deer mice (*Peromyscus maniculatus bairdii*, BW stock, obtained from the Peromyscus Genetic Stock Center at the University of South Carolina), and prairie voles (*Microtus ochrogaster*, obtained from the laboratory of James Burkett at the University of Toledo). Briefly, euthanized rodents were dissected to isolate the testes, from which the caudal epididymis was further isolated. The cauda was moved to a Petri dish containing phosphate-buffered saline (PBS), where it was dissected into small pieces using surgical scissors. The plate was then incubated in a $CO_2$ incubator at 37 °C for 1 h to allow sperm to swim out of the caudal tissue. PBS containing swimming sperm was collected in a 15 mL Falcon tube and centrifuged at $1000 \times g$ for 8 min. The supernatant was discarded, and the pellet was resuspended in an appropriate volume (100–200 μL) of PBS for slide preparation.

Canine semen was obtained from donated ejaculated samples collected manually and cryopreserved with approval from the Cornell University Institutional Animal Care and Use Committee (IACUC). Briefly, following manual collection, samples were diluted 1:1 (v:v) in Irvine Scientific Refrigeration media before being centrifuged at $900 \times g$ for 10 min. The supernatant was discarded, and the sperm pellet was resuspended with refrigeration media to a concentration of 400 million sperm/mL, then diluted with Irvine Scientific Semen Freezing media to a final concentration of 200 million sperm/mL before being loaded into 0.5 mL straws. Samples were cooled to 5 °C for 1 h, then placed in liquid nitrogen vapor for 10 min before being plunged into liquid nitrogen.

### Testes and eye samples
Human testes samples were collected by the University of Toledo Department of Urology with approval from the University of Toledo Institutional Review Board (IRB #0000202366). Rabbit testes samples were provided by Dr. Jie Xu at the University of Michigan. Bovine testes were purchased from Scholl's Slaughterhouse in Blissfield, Michigan. House mouse, rat, and prairie vole testes and eyes were collected from discarded euthanized research animals at the University of Toledo Department of Laboratory Animal Resources and the laboratory of Dr. James Burkett. Deer mouse testes were collected from discarded research animals at the University of Maryland. Animals were bred in accordance with guidelines established by the University of Maryland Institutional Animal Care and Use Committee (protocol R-Jul-18-38).

### Preparation of slides for immunofluorescence studies
Sperm samples were isolated according to the protocol described in the "Caudal sperm isolation" section and were further processed for immunofluorescence studies. Briefly, approximately 20 μL of sperm suspension in PBS buffer were placed on glass slides (Azer Scientific EMS200A+), coverslips (VWR, Cat. 48366-205) were placed, and the slides were dropped into liquid nitrogen. Slides were stored in liquid nitrogen until needed for immunostaining.

### Preparation of testes for confocal and STORM imaging
Pieces of testes samples were embedded in OCT (EMS Diasum 62550-01), and the OCT-embedded testes were then frozen on dry ice and cryo-sectioned. For confocal microscopy, 8-μm sections were made and placed on glass slides. For STORM imaging, 2-μm sections were

made and placed on 18-mm round coverslips. Sections were stored in a −80 °C freezer until needed for immunostaining.

## Antibodies

Primary and secondary antibodies were purchased from various suppliers (Supplementary Table 4). Concentrations of various antibodies used for confocal and STORM microscopy are listed in Supplementary Table 4.

## Immunostaining for confocal and STORM imaging

Immunostaining of sperm and testes for confocal imaging: slides containing sperm samples were removed from liquid nitrogen, coverslips removed using forceps, and the slides were placed in a Coplin jar containing pre-chilled methanol for 3 min. Similarly, slides containing testis sections were removed from the −80 °C freezer and immediately placed in pre-chilled methanol for fixation. Next, the slides were placed in PBS for 1 min, then incubated for 60 min at room temperature in fresh PBS with 0.3% Triton X-1000 (PBST) for permeabilization. Blocking was then performed by placing the slides in 1% PBST-B (1% bovine serum albumin (BSA) in PBST solution) for 1 h. Slides were removed from PBST-B, and primary antibodies diluted in PBST-B were added. The slides were then covered with parafilm tape, placed in a humidity chamber, and incubated overnight at 4 °C. On the following day, the slides were washed three times with PBST for 5 min per wash. Next, the secondary antibody mixture was prepared by adding secondary antibodies and Hoechst (Cat. #H3569, Thermofisher Scientific, 1:200) to a PBST-B solution. The secondary antibody mixture was then added to the slides, which were covered with parafilm and incubated for 3 h at room temperature. Slides were then washed three times with PBST for 5 min per wash, followed by three washes with PBS for 5 min per wash. The slides were then covered with cover glass (18 × 18 mm), sealed using clear nail polish, and stored at 4 °C until needed for imaging.

Immunostaining of testes samples for STORM imaging: immunostaining was performed as described above with slight modifications. Incubation in the primary antibody was done for 24 h at 4 °C, and incubation in the secondary antibody was done for 4 h at room temperature. PBST and PBS washes after primary and secondary antibody incubations, respectively, were performed five times for 5 min per wash. After washing, the cover glasses were stored submerged in PBS at 4 °C. Imaging was performed within 24 to 48 h.

## Confocal imaging

Immunostained slides were imaged using a Leica SP8 confocal microscope. Images were processed using the Leica HyVolution 2 System. Images were captured at a magnification of 63X and a zoom of 1.5X, with a pixel density of 2048 × 2048. Using Adobe Photoshop, confocal images were cropped to 200 × 200 pixels or 65 × 65 pixels. The overall intensities of the images were modified to allow easy visualization, and panels were resized to 1 × 1 inch at 300 DPI for publication.

## 3D STORM Imaging

3D STORM imaging was performed using a Nikon N-STORM 4.0 system on an Eclipse Ti inverted microscope, an Apo TIRF 100X SA NA 1.49 Plan Apo oil objective, 405-, 561-, 488-, and 647-nm excitation laser lines (Agilent), and a back-illuminated EMCCD camera (Andor, DU897). The 647-nm laser line was used to promote fluorophore blinking. Approximately 30,000 time points were acquired at a 20 Hz frame rate, each measuring 16–20 ms in duration. NIS-Elements (Nikon) imaging software was used to analyze and present the data.

Cover glasses were mounted on a depression slide in imaging buffer (10% dextrose in 100 mM Tris at pH 8.0, 25 mM β-Mercaptoethylamine, 0.5 mg/mL glucose oxidase, and 67 μg/mL catalase). Each cover glass was sealed with Body Double SLK (SO56440A and

SO5644B) and allowed to air dry for 3 min, after which the sample was processed for imaging. For imaging various cells in the testes, cells were identified using the phase contrast objective based on their location relative to the basal lamina. Once each cell type was identified, the objective was switched to the TIRF (total internal reflection fluorescence) objective, and each cell was imaged. All STORM imaging for each figure was replicated at least three independent times. Data shown in all figures are cumulative of all replicates.

Z-Calibration was performed using fluorescent beads and stored as a file for each objective and buffer condition according to the manufacturer's instructions. For our STORM analysis, all images were taken using the 100X objective and the same buffer (see above). The raw acquisition data obtained for all STORM images were analyzed using NIS-Elements. Analyzed images were exported as TIFF files. Publication-ready STORM images were prepared in Adobe Photoshop by cropping to 300 × 300, 100 × 100, or 65 × 65 pixels and resizing cropped images to 1 × 1 inch at 300 DPI. The background intensity of the entire image was enhanced for clear visualization of the sperm head.

## Western blot

Testis protein extraction was performed in RIPA buffer with a cocktail of protease inhibitors (Roche cat. 11836170001), which was freshly added before beginning protein extraction. Testes were collected and stored in liquid nitrogen until protein extraction, at which time, the testis sample was placed in ice for 10 min, then minced into small pieces. Approximately 300–500 μL of RIPA buffer was added, and the whole tissue sample was homogenized. The sample was incubated on ice for 1 h, with brief vortexing every 10 min. The sample was then centrifuged at $15,000 \times g$ for 20 min at 4 °C. The supernatant was then collected, loading dye was added, and the mixture was boiled for 10 min. After boiling, the sample was centrifuged again at $15,000 \times g$ for 10 min to sediment any precipitate and to clear any particles from the lysate. The lysate was aliquoted and stored in a −80 °C freezer until needed for western blotting. Each sample was run in either 6% or 8% SDS polyacrylamide gel, then transferred to a PVDF membrane. After transfer, the membrane was blocked in 5% TBSTB (5% BSA in TBST solution), then incubated in primary antibody (see Supplementary Table 4 for antibody concentration) overnight at 4 °C. The membrane was washed three times with TBST for 5 min per wash, then incubated in HRP-conjugated secondary antibody for 1 h at room temperature. Radiance Plus (Azure Biosystem, cat. S1006) was used to detect peroxidase activity. Molecular masses were determined using an EZ-Run Prestained Rec Protein Ladder (Fisher BioReagents, cat. BP3603-500) and an Accu Prestained Protein I Ladder (Lambda Biotech, cat. G01). Western blot imaging was performed using an Azure Imaging System Model 600. Publication-ready western blot images were prepared in Adobe Photoshop.

## Bioinformatic analysis

The longest isoforms of all analyzed proteins were obtained from the NCBI database. Proteins were compared using the NCBI pBLAST tool, and percent identity for FAM161A was extracted. Extended identity ratios were calculated by dividing the sum of house mouse FAM161A percent identities with humans, rabbits, and bovines by the sum of FAM161A percent identities between humans, rabbits, and bovines.

## Intensity measurement by photon counting

For intensity measurements of the various proteins by antibody labeling, images were captured using a confocal microscope in counting mode. All images were captured using a constant laser power of 5%. The images were then analyzed for protein intensity using the Leica LAS X program. Briefly, a round region of interest (ROI) 1.5 μm in diameter was drawn to encompass the proximal centriole and distal centriole, from which pixel sum intensity was recorded and exported

into a Microsoft Excel spreadsheet for additional calculations. For individual centriole intensity measurements, a rectangle ROI was drawn to encompass an individual centriole, and pixel sum intensity was recorded. The measured intensity of all proteins was normalized with the background intensity. Additionally, the intensity data points for POC1B-ab2 and FAM161A in bovine and CETN1 in house mice were normalized by dividing by a factor of five, fifteen, and two, respectively, so that they fit a similar scale.

### Colocalization analysis

For colocalization analysis, a group of transfected cells were randomly selected on a slide, and images were captured using a confocal microscope. An ROI was drawn to surround every transfected cell. The Pearson correlation coefficient of colocalization (r) was measured using the colocalization measurement tool in the Leica LAS X program.

### STORM image quantification

Measurements of proximal and distal centriolar dimensions captured by STORM were performed using Nikon's NIS-Elements imaging software. No STORM images were excluded from quantification. All images were quantified for all parameters as much as possible. To measure proximal and distal centriolar lengths and widths, a line starting at 50% of the first intensity peak through 50% of the last intensity peak was drawn and measured along each length and width.

### Transfection

Transfection of U2OS cells was performed for localization studies. The cells were transfected using GenJet™ In Vitro DNA Transfection Reagent (SignaGen Laboratories, catalog # SL100489-OS) for U2OS Cells as instructed. Briefly, the cells were divided into 6-well plates with round coverslips, 12 h before transfection. On the following day, at least 1 h before transfection, the media was replaced with 1 mL of fresh DMEM complete media. Then, in one set of 1.5 mL microcentrifuge tubes, the DNA mixture was made in 50 μL incomplete DMEM media. Note that a total of 1 μg of DNA was added to make the DNA mixture. In all transfection reactions, an equal amount of DNA was added. In another set of microcentrifuge tubes, the reagent mixture was made by adding 3 μL of transfection reagent to 50 μL incomplete DMEM media. The reagent mixture was then added to the DNA mixture, vortexed briefly, and incubated at room temperature for exactly 15 min. The transfection reaction was then added to the culture, and the plates were shaken well to distribute the reaction mixture uniformly. The transfected cells were then incubated at 37 °C in a 5% $CO_2$ incubator for 48 h. Finally, the transfected cells were processed for immunostaining and confocal imaging.

### Constructs and plasmids

House mouse FAM161A cDNA of type 2 (transcript variant X3, accession number XM_006514830.3) and type 3 (transcript variant 1, accession number NM_001363282.1) was purchased from GenScript (Clone IDs OMu45282 and OMu45285, respectively). FAM161A type 2 was tagged with FLAG, and type 3 was tagged with HA. Both isoforms were cloned into a pcDNA 3.1 mammalian expression vector using an NEBuilder® HiFi DNA Assembly kit (cat # E5520S, New England Biolabs). We obtained human FAM161A cDNA in p3XFLAG-CMV-7 from Dr. Frans P.M. Cremers[109]. pGBKT7 and pGADT7 plasmids for yeast two-hybrid assays were obtained from Dr. Scott Crawley at the University of Toledo. pIC194-POC1B was reported previously (Cekic, 2017). POC5-GFP was obtained from Dr. Michel Bornens. All cloning was performed using either the NEBuilder® HiFi DNA Assembly Cloning kit or the classical digestion and ligation method. Cloning was confirmed by colony PCR, and the accuracy of cloned DNA sequences was confirmed by sequencing.

All interaction mapping was performed using human FAM161A type 2, human POC1B, and human POC5. FAM161A fragments (141–660

and 230–161) were cloned in a p3XFLAG-CMV-7 mammalian expression vector.

### Yeast two-hybrid assay

The GAL4-based yeast two-hybrid system was used to identify binary protein-protein interaction and to construct an interaction map. The constructs encoding full-length FAM161A, POC5, POC1B, and their various fragments were fused to either the DNA binding domain (GAL4-BD) in a pGBKT7 plasmid or the DNA activation domain (GAL4-AD) in a pGADT7 plasmid. Protein interaction was studied by transforming different combinations of GAL4-BD and GAL4-AD DNA constructs in the AH109 yeast strain. Yeast was transformed using the Frozen-EZ Yeast Transformation II Kit (#T2001) from Zymo Research.

Positive transformation of both plasmids was selected for using double dropout media (DDO) lacking Leu and Trp. The interaction was confirmed by growing yeast in media of varying stringency levels and by using triple (TDO, -Leu, -Trp, -His) and quadruple dropout media (QDO, -Leu,-Trp,-His,-Ade).

### FAM161A mRNA amplification and sequencing

mRNA Extraction: Testes and eyes were collected from euthanized house mice and stored in liquid nitrogen until mRNA extraction, which was performed using the Roche mRNA Isolation kit (Cat. No. 11741985001). In summary, -100 mg of snap-frozen tissue was ground to a homogenous powder in a pre-cooled mortar and added to 1.5 mL lysis buffer pre-chilled to 0 °C. The powder suspension was then homogenized by passing 4 times through a 21-gauge needle and centrifuged at $11,000 \times g$ for 30 s, after which the supernatant was transferred to a "sample tube." The following steps were then performed on ice (at a temperature between 0 °C and −4 °C): 150 μL re-suspended streptavidin-coated magnetic particles (SMPs) were pipetted into a fresh tube, then immobilized on the side of the tube using a magnetic particle separator (Permagen® Labware, PN: MSR06). SMP storage buffer was then removed, and SMPs were re-suspended in 250 μL lysis buffer. SMPs were again immobilized with the magnetic particle separator, and all lysis buffer was removed. Next, 1.5 μL biotin-labeled oligo(dT)$_{20}$ probe was added to the sample tube and mixed with the sample to form a "hybridization mix." The hybridization mix was then added to the prepared SMPs, the SMPs were resuspended, and the mixture was incubated on ice for 5 min. Following incubation, SMPs were separated from the mixture with the magnetic particle separator. The SMPs were then resuspended in 250 μL wash buffer and separated from the buffer using the magnetic particle separator, after which all wash buffer was removed and discarded; this series of steps was repeated two more times. SMPs were then re-suspended in 25 μL double-distilled water and incubated for 2 min at 65 °C on a metal heat block. SMPs were then separated from the eluate with the magnetic particle separator, and the supernatant (-25 μL, containing the mRNA) was transferred to a fresh, RNase-free tube.

3' RACE: First-strand cDNA synthesis was performed using the Roche 5'/3' RACE Kit, 2nd Generation (Cat. No. 03353621001). The following were pipetted into a sterile, 0.2-mL Thermowell® Gold PCR Tube (Corning, Inc.) on ice, mixed, and spun down briefly: 4 μL cDNA Synthesis Buffer, 2 μL Deoxynucleotide Mixture, 1 μL Oligo d(T)-Anchor Primer, 5 μL total RNA from the mRNA Extraction step above, 1 μL Transcriptor Reverse Transcriptase, and 7 μL double-distilled water. The mixture was then incubated in a C1000 Touch™ Thermal Cycler (Bio-Rad) for 60 minutes at 55 °C, then for 5 min at 85 °C.

The cDNA was then directly amplified by PCR. The following were pipetted into a sterile microcentrifuge tube on ice, mixed, and spun down briefly: 2 μL cDNA product from the previous step, 2.5 μL primer name 2FW, 2.5 μL PCR Anchor Primer, 1 μL Deoxynucleotide (dNTP) Solution Mix (New England Biolabs, Inc., N0447L), 10 μL Q5® Reaction Buffer (New England BioLabs, Inc., B9027SVIAL), 10 μL Q5® High GC Enhancer (New England BioLabs, Inc., B9028AVIAL), 0.5 μL Q5® High-

Fidelity DNA Polymerase (New England BioLabs, Inc., M0491LVIAL), and 21.5 µL double-distilled water. The reaction mix was placed in a C1000 Touch™ Thermal Cycler, and PCR was run with the following conditions: initial denaturation at 98 °C for 30 s; 35 cycles of denaturation at 98 °C for 10 s, annealing at 55 °C for 30 s, and extension at 72 °C for 2.5 min; final extension at 72 °C for 2 min; and indefinite hold at 4 °C. 10 µL of the first PCR amplification product was then used for analysis on a 2% ethidium bromide-stained agarose gel (120 min at 80 V in a Mini-Sub® Cell GT gel chamber connected to a PowerPac™ HC (Bio-Rad) power supply) with a 1kB Plus DNA Ladder (New England BioLabs, Inc., N3200L) as molecular weight marker.

For the second PCR amplification, the following were pipetted into a sterile, 0.2-mL Thermowell® Gold PCR Tube on ice, mixed, and spun down briefly: 2 µL first PCR amplification product, 2.5 µL primer name 3FW, 2.5 µL primer name 5RV, 5*RV, or 7RV, 1 µL dNTP Solution Mix, 10 µL Q5® Reaction Buffer, 10 µL Q5® High GC Enhancer, 0.5 µL Q5® High-Fidelity DNA Polymerase, and 21.5 µL double-distilled water. The reaction mix was placed in a C1000 Touch™ Thermal Cycler, and PCR was run with the following conditions: initial denaturation at 98 °C for 30 s; 35 cycles of denaturation at 98 °C for 10 s, annealing at 58 °C for 30 s, and extension at 68 °C for 2.5 min; final extension at 72 °C for 2 min; and indefinite hold at 4 °C. All 50 µL of the second PCR amplification product was then used for analysis on a 2% ethidium bromide-stained agarose gel (120 min at 80 V in a Mini-Sub® Cell GT gel chamber connected to a PowerPac™ HC power supply) with a 1kB Plus DNA Ladder as molecular weight marker. DNA bands were visualized and imaged using a DIGI DOC-IT High Performance Ultraviolet Transilluminator (UVP, P/N 97-0105-01), then excised and placed individually in 2-mL centrifuge tubes (G-Biosciences, Cat. #C041-D) for DNA purification.

DNA Purification: DNA purification was performed using the QIAquick® Gel Extraction Kit (cat. Nos 28704 and 28706) with slight modifications. In summary, 700 µL Buffer QG was added to the tube containing an excised agarose gel slice. The tube was then incubated for at least 10 min at 50 °C on a metal heat block and vortexed every 2 to 3 min until the gel slice was completely dissolved. Then, 350 µL isopropanol was added to the tube and mixed. Approximately 525 µL of the sample was then applied to a QIAquick® spin column seated in a provided 2 mL collection tube and centrifuged for 1 min at 15,000 RPM. The flow-through was discarded, the QIAquick® column was placed back into the same collection tube, and the loading and spin were repeated with the remaining volume of the sample, followed by placement of the QIAquick® column back into the same collection tube. To wash the sample, 750 µL Buffer PE (with ethanol added) was added to the QIAquick® column and allowed to stand for 5 min at room temperature, then centrifuged for 1 min at 15,000 RPM. The flow-through was discarded, the QIAquick® column was placed back into the same collection tube, and the column was centrifuged for an additional 2 min at 15,000 RPM to remove residual wash buffer. The column was then placed into a clean, 1.5-mL microcentrifuge tube. To elute DNA, 15 µL sterile water warmed to 65 °C on a metal heat block was added to the center of the QIAquick® membrane and incubated for 5 min at room temperature, then centrifuged for 2 min at 15,000 RPM. To increase the yield of purified DNA, the previous elution steps were repeated. Purified DNA was then prepared for sequencing.

DNA Sequencing: Purified DNA was prepared for sequencing by pipetting the following into a sterile, 0.2-mL Thermowell® Gold PCR Tube (Corning, Inc.): 2 µL purified DNA, 2.5 µL sequencing primer (primer name Seq. 3FW, Seq. 5RV, Seq. 5*RV or Seq. 7RV), 10.5 µL sterile water (see Supplementary Table 5 for primer information). Sequencing was performed by GENEWIZ (Azenta Life Sciences), and sequencing results were analyzed using Sequencher 5.2.2 (Gene Codes Corporation).

## Statistical analysis and reproducibility

Unless otherwise noted, each experiment was independently performed at least three times with similar results from at least three distinct samples. All averages and standard deviations in this study were calculated using Microsoft Excel. All correlations, regressions, and $t$-test analyses were performed using GraphPad Prism 8.0. In each figure or figure legend, the number (n) of cells analyzed and all $P$-values are specified. An unpaired, two-tailed $t$-test was used to perform the statistical analysis in this study; normality was assumed. All quantification data are presented as box and whisker and scatter plots. All box and whisker plots are represented as minimum to maximum, showing all data points, medians, and interquartile ranges. Each data point in all scatter plots represents a measurement for an individual cell. $P$-values are indicated as asterisks (*) denoting the significance of comparison: $*P < 0.05$, $**P < 0.01$, $***P < 0.001$, $****P < 0.0001$.

## Estimation of ω (dN/dS) across centriolar proteins

Codon alignments and phylogenetic tree topologies of FAM161a, FAM161b, CETN1, POC5, POC1b, and WDR90 were used for the estimation of ω across the gene. We employed the CODEML null model (M0) of the PAML suite to estimate ω across codon alignments. We estimated ω for FAM161a orthologs from Rodents, Carnivores, Ungulates, and Primates using the respective coding DNA sequence (CDS) alignments. We also estimated ω in individual subgroups of Rodents (Muridae, Cricetidae and other myomorpha species).

## Identification of sites under positive and negative selection within rodent groups

We used several of the tests for identifying sites under selection available through the HyPhy suite. The Mixed Effects Model of Evolution identifies sites under episodic diversifying/positive selection, whereas FUBAR and FEL identify sites under pervasive positive/negative selection across codon alignments.

## Ortholog search

We extracted FAM161A orthologs for rodents, ungulates, carnivores, and primates from the NCBI Ortholog list. The longest mRNA transcript sequence of the longest isoform (isoform X1) was selected for all species analyzed (Supplementary Table 6). We used house mouse FAM161A as a query and performed a tBLASTn search against rodent taxa to identify orthologs missing from the NCBI ortholog list (last accessed). We used the AUGUSTUS gene prediction tool to identify the CDS of all FAM161A orthologs to ensure the correct coding frame of the gene. We gathered a total of 115 orthologs from four mammalian groups.

## Sequence alignment and phylogenetic reconstructions

The CDS of individual animal groups were aligned using MUSCLE[132]. The CDS was translated into its amino acid sequence before alignment and reverse translated to recover the aligned codon positions. Alignments were manually curated by removing gaps and stop codons. Phylogenetic trees were constructed by employing a Bayesian phylogenetic framework. The model of sequence evolution was determined using the ModelFinder program in IQTree[133]. Phylogenetic reconstruction was performed using MrBayes[134] with GTR + I + G as the model of sequence evolution. We parallelized four Monte Carlo chain simulation runs to execute the analysis and terminated the analysis when the standard deviation of split frequency reached less than 0.01. The models and tree were sampled every 100th generation, and after the simulation was complete, the initial 25% of the sampled trees and their corresponding parameters were discarded as burn-in. Fine tree topography and node support values were estimated for the tree data that was retained after excluding burn-in. The tree was visualized using FigTree[135].

## Selection analysis

We implemented maximum likelihood-based selection models in CodeML binaries within the PAML package[136] to test the nature of selection acting on FAM161A orthologs in different species groups. The ratio of non-synonymous substitutions (nucleotide substitutions that change the protein sequence, denoted dN) to synonymous substitutions (nucleotide substitutions that do not change the protein sequence, denoted dS), known as 'ω' (dN/dS), was estimated across codon positions to identify the signature of selection. We compared the likelihoods of two pairs of models, M7 (null model) vs. M8 (alternate model) and M8a (null model) vs. M8 (alternate model). M7 and M8a are neutral models, whereas M8 is a positive selection model. The significance of the difference between the likelihood values of both models was compared using chi$^2$ distribution. Amino acid sites evolving under the influence of positive selection ($\omega > 1$) were identified using the Bayes Empirical Bayes (BEB) by M8 selection model. Furthermore, we used MEME to identify the episodic effect of natural selection and FEL and FUBAR to identify the sites under pervasive effect of selection.

We further applied branch-site models in PAML to check the lineage-specific effect of positive selection in the Muridae and Cricetidae branches of rodent phylogeny. We ran M1 and M1a models and compared the significance of the difference between likelihood values using a chi$^2$ test.

## Reporting summary

Further information on research design is available in the Nature Portfolio Reporting Summary linked to this article.

## Data availability

Most relevant data supporting the findings of this study are available in this manuscript and in the supporting information file. Additional data can be found under the project title "The Evolution of Centriole Degradation in Mouse Sperm" in the Figshare public repository. Finally, any additional data not shown in this manuscript are available from the corresponding author upon request. Source data are provided with this paper.

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

## Acknowledgements

We would like to thank the University of Toledo Department of Laboratory Animal Resources (DLAR) and Drs. Heather Conti and Fan Dong at the University of Toledo for rat and house mouse samples. Testis sections were made by Allen Schroering, University of Toledo Medical Center (UTMC), Ohio. FAM161A N-terminus antibody was obtained from Dr. Dror Sharon, Hadassah-Hebrew University Medical Center. POC5 antibody was gifted by Dr. Michel Bornens. Plasmids for yeast two-hybrid analyses were obtained from Dr. Scott Crawley, University of Toledo, Ohio. We thank Dr. Bo Harstine at Select Sires, Inc. for donating bovine (Holstein) spermatozoa straws. We thank Dr. Kristin Kirshenbaum and the NSM Instrumentation Center at the University of Toledo for the use of the STORM microscope. We thank Professor Geoff Parker (University of Liverpool) for commenting on the paper. We thank Dr. Jie Xu for help with the rabbit samples. We thank Rose Soriano for help with the bioinformatic analysis. This project was supported by Agriculture and Food Research Initiative Competitive Grant no. OHOW-2020-02790 from the USDA National Institute of Food and Agriculture and by NIH-NICHD 1R15HD110863.

## Author contributions

All authors read, edited, and approved the paper for publication. S.K. Led the project and extensively edited the paper. A.J. Performed tissue culture cell studies. R.C. Performed the molecular phylogeny analysis, draft edit. N.P. Performed RT-PCR studies and extensively edited the paper. K.Y.A. Assisted with molecular biology. K.T. Performed initial experiments with rabbits, draft edit. M.N. Assisted with domain analysis. E.B. Assisted with western blots. A.R. Assisted with STORM. J.B. Provided access to *Microtus ochrogaster* sperm and testes. S.H.C. Provided access to dog sperm. H.S.F. Provided access to *Peromyscus* sperm and testes. P.S. Provided access to human sperm and testes. J.G. RNA-seq analysis. R.N.B. Oversaw molecular phylogeny analysis, draft edit. T.A.R. Led the project and wrote the first draft.

## Competing interests

The authors declare no competing interests.
