## [Peer Review File · Nature Communications]

The Evolution of Centriole Degradation in Mouse SpermReviewer #1 (Remarks to the Author):

Review of Khanal et al. MS, "Centriole Remodeling Evolved into Centriole Degradation in Mouse Sperm."

This manuscript is somewhat unusual in that it is in some senses two manuscripts in one: an extensive literature review on the subject of sperm centriole structure and evolution of a specific centriolar protein (FAM161) across many species, combined with a relatively small set of fluorescence images supporting some of the ideas generated by that review.

Both are valuable contributions to the literature and of interest primarily to specialists in the area of sperm structure function and evolution. There may be broader implications for the general topics of organelle evolution or of cilium evolution more generally, but the MS makes little effort to draw attention to those.

A few suggestions:

1) Consider separating into two papers.

2) Change the title. It is objectionable to have as title a declaratory sentence about evolution. Evolutionary theories may be consistent with existing data or not and have useful predictive power or not, but cannot be determined to have been conclusively demonstrated to be "correct" descriptions of how evolution occurred over millions of years and countless numbers of individual organisms.

3) Provide some figures (perhaps in Supplementary Material) of some samples of the published electron microscopy data on which the section summarizing these data are based. It is very difficult (not to mention dry and boring) to read through extensive verbal descriptions of three dimensional structures without seeing the data on which those are based. The highly simplified schematic diagrams provided are not particularly useful in this regard.

4. Some discussion of how eggs may have evolved alongside major changes in sperm centriole structure would be appropriate, as well as a clarification of what is meant by "centriole-independent development." Clearly in some cases sperm centrioles have become dispensable for development, but obviously the corresponding embryos cannot progress without centrioles; the interesting question is their source. Any images documenting embryonic centriole/centrosome/ciliary dynamics in those species would be of interest.

5. Experimental:

Once they discovered that Type 3 isoform of FAM161A is expressed in the mouse sperm, did they make/find an antibody specific for that? This could be critical since they see that the type 2 isoform is still expressed, and they compare the localization pattern of fam161A between species and between sperm cell development stages. If such critical experiments are not feasible the reasons and implications should be discussed.

The STORM image is nice, but it seems odd that there is only one, although optimization of the corresponding protocol is described. It would be nice to see the difference in centriole protein localization with STORM, if such data exist or could easily be collected. It would be nice to show the localization of multiple centriole proteins with STORM in at least two differing species.

What about Spiroliin protein? (Hum Reprod. 2010 Aug;25(8):1884-94. doi:

10.1093/humrep/deq138. Epub 2010 Jun 11.), which the authors also mention in a review they wrote in 2015. Do they have any evolutionary analysis or relevant data on this candidate in centriole evolution?

In their methods: they say "testis and eye collection" but they have no mention of any eye collection in that section.

Figures:

They should explain in their illustrations what the black line is in the middle of their sperm head nuclei. Since this paper has a broad impact and interest, not only researchers in the sperm field will be reading it, and to someone unfamiliar it looks like that is where the "center" attachment might be and could be confusing. Also, in the supplemental figure 1 illustration they show a red line surrounding part of the nucleus head, and they should explain what that is as well.

For the fluorescence figures it is a bad idea to use red and magenta together. Alternative color schemes, preferably those friendly to the large group of color-blind readers, should be judiciously applied <https://www.ascb.org/science-news/how-to-make-scientific-figures-accessible-to-readers-with-color-blindness/>.

Reviewer #2 (Remarks to the Author):

The centrioles found in sperm vary across species and in some rodent species are entirely degraded. Here, Khanal et al. address the important and interesting question of how Muridae species evolved to degrade sperm centrioles. The authors show a correlation between the evolution of a lateral head-neck connection and centriole degradation. They further show that evolution of centriole loss is correlated with structural changes in the centriolar luminal protein FAM161A, and that localization of FAM161A in house mouse sperm is unique compared to other mammalian species. As relatively little is known about how and why sperm centrioles are degraded in some species, this paper addresses an important knowledge gap and generates an exciting hypothesis that centriole degradation evolved to facilitate lateral head-tail attachments. While I do not think additional experiments are needed, I recommend that the language used throughout the text must be more precise, as certain claims are overstated or misleading.

Major comments:

1. The authors have described an analysis workflow that led to the discovery of FAM161A as a potential differentially regulated protein with various isoforms performing different functions. While the data does allow the reviewers to distinguish localization differences, the leap to a functional difference is quite a stretch. It is definitely beyond this study to investigate function and causality. I recommend the editor insist that any difference in function clearly be stated as a hypothesis, and that FAM161A only correlates with centriole structure with no cause and effect tested. There is no insight into the actual role of microtubule localization, the nuclear localization, or centriole levels. Examples:

- Line 151-2: This is an overstatement and should be rephrased as a correlation or hypothesis.
 - Line 364: the authors state that "FAM161A has distinct functions in house mouse and human sperm." This is misleading as these experiments were performed in U2OS cells. Further, it is a stretch to say that there are functional differences between human and mouse FAM161A and between mouse FAM161A type2 and type3 when only localization differences are shown. If FAM161A is a centriolar protein, why is it accumulating in foci in the nucleus?
 - Line 391: the localization pattern does not "indicate," it merely suggests or allows for the hypothesis that functional difference exist.
 - Lines 565 and 577: the use of the term "association" suggests causation, it should be changed to correlation.
 - Line 568: This stamen is exceptionally strong. In no way did this study "show."
- Changes to the language to better reflect that these are correlations and hypotheses would address this major concern.

2. The measurement of FAM161A levels on centrioles has led to the hypothesis that there is a regulated molecular mechanism that distinguishes the house mouse from others. Specifically, the ES column of FAM161A in Figure 7D. Is this measurement a total of all FAM161A? Have the authors measured protein levels in deer mouse, bovine, rabbit, humans in the proximal vs distal centriole separately? Does the proximal and distal centriole recruit proteins in the same pattern? If they are different, do either centrioles look more similar to the single dot measured in house mouse? The results of separating the data could change the model.

Additional comments:

1. A critical part of this study is the survey of past literature. While the tree in figure 1B describes what they saw in published TEM studies, it is somewhat a shame that the authors didn't share these images in this manuscript. I recognize that copyright issues might be at play here, but such a compiled collection of past images would be a valuable resource to the field.
2. Given that FAM161A was not the top hit from their protein sequence alignment comparison, the authors should show the top 10 hits with the lowest Identity Ratio in the main figure 2. In addition, EIR should be calculated for these top hits, not just for FAM161A.

Minor comments

1. A significant number of figure citations are incorrectly labeled. Please ensure that figure citations match what is actually shown in the figure.

2. Grayscale should be used for single channel images, especially in figures 5, 6, and 7.
3. It is also not clear if data shown is compiled from multiple experimental replicates or representative replicates. Please clearly indicate Ns of centrioles, cells, experiments in all legends.
4. Please specify which FAM161A isoform is shown in Figure 3G.
5. Arrows to point out "v" structures in Figures 6 and 7. Arrows to point out "splayed" centrioles in Sfig6-1
6. It would be helpful to highlight the 95aa region with unknown function that is missing in FAM161A type3 in SFig4-2.
7. On line 370, the authors state they "mapped binding between human FAM161A type 2 and the other 3 human rod proteins" but mapping is only shown for POC5 and POC1B.
8. In many areas throughout the manuscript the authors fail to reference either literature or their own data. For example: Line 421-423 – at the of the sentence, the authors should indicate a figure or a reference, even if the same figure was reference the previous sentence. Another example is 172-173, 362-363, 477-478. All of these leave the reading asking if "who showed this?"
9. Centriole adjunct is never defined.

Reviewer #3 (Remarks to the Author):

The work of Khanal et al. is based on the hypothesis that sperm centrioles have evolved through a cascade of evolutionary changes in the proteins composing the centriole. To test it, they performed phylogenetic analyses and discovered a correlation between the loss of the centriole structure and a change in the protein sequence of the protein FAM161, a microtubule-associated protein lying inside the centriole. The authors also found that the sequence of FAM161A in Muridae et Cricetidae greatly evolved and correlate this with the emergence of centriole-independent embryo development in house mouse. The authors also found a novel FAM161A type 3 isoform in Muridae testes and using cellular assays, they demonstrate that this isoform inhibits the type 2 isoform (the most common found in cells) by preventing its capacity to interact with microtubules. Finally, the authors show that FAM161A is only present in the distal centriole of Cricetidae and Muridae spermatozoan. By analyzing the localization of the protein Centrin, POC5, POC1B, and FAM161A in the three sperm cell stages of humans, rabbits, and bovines, the authors also show the species have two conserved remodeling steps.

Overall the work appeared to be of very good quality and provides new evolutionary insight into the differences in sperm cell structure between species. The work has importance for the field and related fields. However, I am not an expert in evolution, but it would seem that many of the conclusions are correlations without a demonstration of causality. As I am not a specialist in evolutionary and phylogenetic aspects, my review focuses mainly on the cell biology parts:

- Could the authors use a cell-based assay to better understand the effect of the different isoforms? FAM161A is a centriolar protein that is part of the inner scaffold and therefore could test the functions at centrioles, not only in sperm cells. In the experiments presented, the authors show that the interaction with cytoplasmic microtubules disappears in the presence of Type 3, inhibiting Type 2. And it would also be interesting to do a siFAM161A and try to rescue with type 3. The prediction here is that the centriole structure would not be rescued and would generate broken/remnant-like centrioles.
- Similarly, about the localization of the different isoforms, it would be interesting to know if the isoforms also localize to centrioles/centrosomes, can the authors provide these quantifications? Similarly for the interactors, would overexpression of type 3 for more than 48h prevent POC5 from localizing to the centriole? And Wouldn't a mouse cell culture model be more adequate to overexpress mouse proteins?
- In Figure 3B, it appears that there is a second band, potentially FAM161A type 2. Can the authors provide a more exposed blot ?
- Regarding the presence of isoforms, in addition to WBs, it would be important to demonstrate the presence of the protein isoforms using mass spectrometry analysis (with characteristic

peptides of each form).

- The authors use an Alphafold structure prediction to show positively selected sites and two apparent primary domains: core and outer. How confident is this model? The structure prediction of FAM161A is mostly unfolded, right? If the prediction does not have a high score, this type of modeling is misleading and incorrect.

REVIEWER COMMENTS

Answer: We sincerely appreciate the time and effort the reviewers dedicated to providing us with detailed reviews of our paper. We have carefully addressed all the points raised and believe that the paper has undergone significant improvements.

Reviewer #1 (Remarks to the Author):

Review of Khanal et al. MS, "Centriole Remodeling Evolved into Centriole Degradation in Mouse Sperm."

This manuscript is somewhat unusual in that it is in some senses two manuscripts in one: an extensive literature review on the subject of sperm centriole structure and evolution of a specific centriolar protein (FAM161) across many species, combined with a relatively small set of fluorescence images supporting some the ideas generated by that review. Both are valuable contributions to the literature and of interest primarily to specialists in the area of sperm structure function and evolution. There may be broader implications for the general topics of organelle evolution or of cilium evolution more generally, but the MS makes little effort to draw attention to those.

Answer: We appreciate the reviewer recognizing the contribution of our paper to the area of sperm structure function and evolution. As suggested, we added general implications to cilium and organelle evolution in the discussion and expanded fluorescence images supporting the paper's conclusions.

A few suggestions:

1) Consider separating into two papers.

Answer: As suggested by Nature communication editors, we plan to keep the paper as one unit.

2) Change the title. It is objectionable to have as title a declaratory sentence about evolution. Evolutionary theories may be consistent with existing data or not and have useful predictive power or not, but cannot be determined to have been conclusively demonstrated to be "correct" descriptions of how evolution occurred over millions of years and countless numbers of individual organisms.

Answer: We accept this concern. We changed the title, so it has no declaration to *The Evolution of Centriole Degradation in Mouse Sperm*.

3) Provide some figures (perhaps in Supplementary Material) of some samples of the published electron microscopy data on which the section summarizing these data are based. It is very difficult (not to mention dry and boring) to read through extensive verbal descriptions of three dimensional structures without seeing the data on which those are based. The highly simplified schematic diagrams provided are not particularly useful in this regard.

(From the Editor: I agree that compiling the EM images could be of value to the community, especially given your comprehensive comparative descriptions. However, we definitely understand the hurdle and effort required to get the permissions for the published images. I propose that you gather the images into a figure and include it in the revision. If we decide to publish your paper and include that figure in the final version, my Editorial Assistant can help collect the permissions at that stage.)

Answer: We have gathered the images related to Figure 1B and will get permission to publish once the paper is accepted.

4. Some discussion of how eggs may have evolved alongside major changes in sperm centriole structure would be appropriate, as well as a clarification of what is meant by “centriole-independent development.” Clearly in some cases sperm centrioles have become dispensable for development, but obviously the corresponding embryos cannot progress without centrioles; the interesting question is their source. Any images documenting embryonic centriole/centrosome/ciliary dynamics in those species would be of interest.

Answer: We thank the reviewer for these comments. First, we added a discussion about how the maternal mechanisms from eggs may have evolved alongside major changes in sperm centriole. Second, to clarify the “centriole-independent development” phrase, we replaced it with “centriole-independent early development” to indicate that centrioles are not needed only in the early embryo, at the zygote and morula stages. Third, we added a discussion on centriole appearance in the mouse embryo.

5. Experimental:

- Once they discovered that Type 3 isoform of FAM161A is expressed in the mouse sperm, did they make/find an antibody specific for that? This could be critical since they see that the type 2 isoform is still expressed, and they compare the localization pattern of fam161A between species and between sperm cell development stages. If such critical experiments are not feasible the reasons and implications should be discussed.

Answer: Thank you for this suggestion, However, Type 3 has only 5 unique amino acids that are not found in type 1 or 2. Therefore, it would be extremely challenging to create an antibody specific to Type 3.

- The STORM image is nice, but it seems odd that there is only one, although optimization of the corresponding protocol is described. It would be nice to see the difference in centriole protein localization with STORM if such data exist or could easily be collected. It would be nice to show the localization of multiple centriole proteins with STORM in at least two differing species.

Answer: We apologize for the incorrect figure citation in the paper that caused this confusion, as there is more than one image of STORM in the paper. There are 11 images in the paper, and many more are used for the quantification. First, we show nine images in SFig 6-1A, imaging bovine, CETN1, POC5, and POC1B in spermatogonia/spermatocytes and round spermatids (this figure was not cited correctly). Second, we show two mice POC5 images in Fig 7E of STORM, imaging house mouse POC5 in spermatogonia/spermatocytes and round spermatids. Finally, we added mice CETN1 images to the differentiation stages in Fig 7E. We also better indicated which data is based on STORM analysis.

- What about **Speriolin** protein? (Hum Reprod. 2010 Aug;25(8):1884-94. doi: 10.1093/humrep/deq138. Epub 2010 June 11.), which the authors also mention in a review they wrote in 2015. Do they have any evolutionary analysis or relevant data on this candidate in centriole evolution?

Answer: We have looked at Speriolin (SPATC1), which has an identity ratio of 0.95. This information is in STable 02-1 of the original submission.

In their methods: they say "testis and eye collection," but they have no mention of any eye collection in that section.

Answer: Eyes were collected from the same animals used for testis collection. This has been added to the method section now.

Figures:

- They should explain in their illustrations what the black line is in the middle of their sperm head nuclei. Since this paper has a broad impact and interest, not only researchers in the sperm field will be reading it, and to someone unfamiliar it looks like that is where the "center" attachment might be and could be confusing. Also, in the supplemental figure 1 illustration they show a red line surrounding part of the nucleus head, and they should explain what that is as well.

Answer: Thank you for these observations. First, we now deleted the black line in the middle of their sperm head nuclei in Fig 1A. Second, we added an explanation of the red line surrounding part of the nucleus head (as well as the other colors) in the supplemental Figure 1 illustration. We also modified panels Fig 1Aiii and 1Aiv to be consistent with the description in the text. Finally, we changed the graphical abstract accordingly and deleted its black line.

For the fluorescence figures, it is a bad idea to use red and magenta together. Alternative color schemes, preferably those friendly to the large group of color-blind readers, should be judiciously applied <https://www.ascb.org/science-news/how-to-make-scientific-figures-accessible-to-readers-with-color-blindness/>.

Answer: Thank you for this comment. The link you submitted suggests combining Magenta, Green, and Blue, as we do in our paper. However, we have an additional 4th color, we needed to use red for that. Also, we made new identical supplemental figure 6 and 7 that has a single channel in a grayscale.

Reviewer #2 (Remarks to the Author):

The centrioles found in sperm vary across species, and in some rodent species, are entirely degraded. Here, Khanal et al. address the important and interesting question of how Muridae species evolved to degrade sperm centrioles. The authors show a correlation between the evolution of a lateral head-neck connection and centriole degradation. They further show that evolution of centriole loss is correlated with structural changes in the centriolar luminal protein FAM161A, and that localization of FAM161A in house mouse sperm is unique compared to other mammalian species. As relatively little is known about how and why sperm centrioles are degraded in some species, this paper addresses an important knowledge gap and generates an exciting hypothesis that centriole degradation evolved to facilitate lateral head-tail attachments. While I do not think additional experiments are needed, I recommend that the language used throughout the text must be more precise, as certain claims are overstated or misleading.

Answer: We appreciate the reviewer recognizes that this paper addresses an important knowledge gap and generates an exciting hypothesis. As suggested, we made the language used throughout the text more precise.

Major comments:

1. The authors have described an analysis workflow that led to the discovery of FAM161A as a potential differentially regulated protein with various isoforms performing different functions. While the data does allow the reviewers to distinguish localization differences, the leap to a functional difference is quite a stretch. It is definitely beyond this study to investigate function and causality. I recommend the editor insist that any difference in function clearly be stated as a hypothesis, and that FAM161A only correlates with centriole structure with no cause and effect tested. There is no insight into the actual role of microtubule localization, the nuclear localization, or centriole levels. Examples:

- Line 151-2: This is an overstatement and should be rephrased as a correlation or hypothesis.
- Line 364: the authors state that "FAM161A has distinct functions in house mouse and human sperm." This is misleading as these experiments were performed in U2OS cells. Further, it is a stretch to say that there are functional differences between human and mouse FAM161A and between mouse FAM161A type2 and type3 when only localization differences are shown. If FAM161A is a centriolar protein, why is it accumulating in foci in the nucleus?
- Line 391: the localization pattern does not "indicate," it merely suggests or allows for the hypothesis that functional difference exist.
- Lines 565 and 577: the use of the term "association" suggests causation, it should be changed to correlation.
- Line 568: This stamen is exceptionally strong. In no way did this study "show."

Changes to the language to better reflect that these are correlations and hypotheses would address this significant concern.

Answer: We appreciate the reviewer's comments and made changes to all these examples and other locations we noticed in the text. Note that we have added many more experiments to address the localization pattern differences of the various FAM161A isoforms (See Figure 4 and the supplementary figures related to Figure 4).

2. The measurement of FAM161A levels on centrioles has led to the hypothesis that there is a regulated molecular mechanism that distinguishes the house mouse from others. Specifically, the ES column of FAM161A in Figure 7D. Is this measurement a total of all FAM161A? Have the authors measured protein levels in deer mouse, bovine, rabbit, humans in the proximal vs distal centriole separately? Does the proximal and distal centriole recruit proteins in the same pattern? If they are different, do either centrioles look more similar to the single dot measured in house mouse? The results of separating the data could change the model.

Answer: The measurement in Figure 7D is a total of FAM161A in both the PC and DC. We have now added this information to the figure legend. We also added the protein levels in bovines and house mice in the proximal versus distal centriole separately in supplemental figures 6-3 and 7-2. We

found that individual centrioles have similar localization intensity changes to what is observed in the total centriole quantification. We specifically looked at these two organisms because the proximal and distal centriole in the other species could not be reliably distinguished in the round spermatids. Bovines have the easiest to identify proximal and distal centrioles in the round and elongated; additionally, the overall recruitment of centriolar proteins is similar in bovines, humans, and rabbits. We compared bovine's proximal and distal centriole localization intensity to house mice because house mice have the most extreme centriole degeneration. Bovine and house mice are also among the most heavily studied organisms. Since no difference was observed in these species' individual centriole localization intensities, we did not include other species.

Additional comments:

1. A critical part of this study is the survey of past literature. While the tree in figure 1B describes what they saw in published TEM studies, it is somewhat a shame that the authors didn't share these images in this manuscript. I recognize that copyright issues might be at play here, but such a compiled collection of past images would be a valuable resource to the field.

(From the Editor: I agree that compiling the EM images could be of value to the community, especially given your comprehensive comparative descriptions. However, we definitely understand the hurdle and effort required to get the permissions for the published images. I propose that you gather the images into a figure and include it in the revision. If we decide to publish your paper and include that figure in the final version, my Editorial Assistant can help collect the permissions at that stage.)

Answer: We have gathered the images related to Figure 1B and will get permission to publish once the paper is accepted.

2. Given that FAM161A was not the top hit from their protein sequence alignment comparison, the authors should show the top 10 hits with the lowest Identity Ratio in the main figure 2. In addition, EIR should be calculated for these top hits, not just for FAM161A.

Answer: As suggested, we added this information in Figure 2

Minor comments

1. A significant number of figure citations are incorrectly labeled. Please ensure that figure citations match what is actually shown in the figure.

Answer: We apologize for this mistake, and we have corrected the figure citations.

2. Grayscale should be used for single channel images, especially in figures 5, 6, and 7.

Response: We added accessible color schemes with grayscale pictures to panels with the single color channel as supplementary data named "Supplementary information - figure with grayscale single panels."

3. It is also not clear if data shown is compiled from multiple experimental replicates or representative replicates. Please clearly indicate Ns of centrioles, cells, experiments in all legends.

Answer: We apologize for not clarifying that the N number was in the figures, and we added an explanation for it in the legend. The number of experiment repetitions has been added to the legends of all figures.

4. Please specify which FAM161A isoform is shown in Figure 3G.

Answer: Due to the low confidence of this prediction, we deleted Figure 3G.

5. Arrows to point out “v” structures in Figures 6 and 7. Arrows to point out “splayed” centrioles in Sfig6-1.

Answer: White arrows were added to mark the “V” shape and splayed centriole in Figures 6 and Sfig6-1. Please note that “v” structures were not reported in Figures 7.

6. It would be helpful to highlight the 95aa region with unknown function that is missing in FAM161A type3 in SFig4-2.

Answer: As suggested, we highlighted the Uniquely missing amino acids in Type 3.

7. On line 370, the authors state they “mapped binding between human FAM161A type 2 and the other 3 human rod proteins” but mapping is only shown for POC5 and POC1B.

Answer: Thank you, we corrected the sentence as pointed out.

8. In many areas throughout the manuscript the authors fail to reference either literature or their own data. For example: Line 421-423 – at the of the sentence, the authors should indicate a figure or a reference, even if the same figure was reference the previous sentence. Another example is 172-173, 362-363, 477-478. All of these leave the reading asking if “who showed this?”

Answer: We apologize and added fig references to these lines, except that lines 362-363 do not have a figure, and that data is in the sentence itself.

9. Centriole adjunct is never defined.

Answer: Thank you, we added a definition.

Reviewer #3 (Remarks to the Author):

- The work of Khanal et al. is based on the hypothesis that sperm centrioles have evolved through a cascade of evolutionary changes in the proteins composing the centriole. To test it, they performed phylogenetic analyses and discovered a correlation between the loss of the centriole structure and a change in the protein sequence of the protein FAM161, a microtubule-associated protein lying inside the centriole. The authors also found that the sequence of FAM161A in Muridae et Cricetidae greatly evolved and correlate this with the emergence of centriole-independent embryo development in house mouse. The authors also found a novel FAM161A type 3 isoform in Muridae testes and using cellular assays, they demonstrate that this isoform inhibits the type 2 isoform (the most common found in cells) by preventing its capacity to interact with microtubules. Finally, the authors show that FAM161A is only present in the distal centriole of Cricetidae and Muridae spermatozoan. By analyzing the localization of the protein Centrin, POC5, POC1B, and FAM161A in the three sperm cell stages of humans, rabbits, and bovines, the authors also show the species have two conserved remodeling steps.

Answer: Thank you for summarizing our findings.

- Overall the work appeared to be of very good quality and provides new evolutionary insight into the differences in sperm cell structure between species. The work has importance for the field and related fields. However, I am not an expert in evolution, but it would seem that many of the conclusions are correlations without a demonstration of causality. As I am not a specialist in evolutionary and phylogenetic aspects, my review focuses mainly on the cell biology parts:

Answer: Thank you for recognizing the quality of our work. As was indicated by the other reviews, it is very challenging to demonstrate causality in evolutionary studies, and therefore changed the wording in the paper to be more careful with our conclusions.

- Could the authors use a cell-based assay to better understand the effect of the different isoforms? FAM161A is a centriolar protein that is part of the inner scaffold and therefore, could test the functions at centrioles, not only in sperm cells. In the experiments presented, the authors show that the interaction with cytoplasmic microtubules disappears in the presence of Type 3, inhibiting Type 2. And it would also be interesting to do a siFAM161A and try to rescue with type 3. The prediction is that the centriole structure would not be rescued and would generate broken/remnant-like centrioles.

Answer: Unfortunately, the knockdown of FAM161A using RNAi in U20S cells has consequences only in a minority of the cells, limiting the value of this experiment. Please see "While FAM161A was absent in 88% of centrioles in FAM161A-depleted cells, only 38% of centrioles had lost POC5 (S8 Fig), suggesting that FAM161A does not control POC5 recruitment at centrioles" in Mercey, Olivier et al. "The connecting cilium inner scaffold provides a structural foundation that protects against retinal degeneration." *Plos Biology* 20.6 (2022): e3001649. In this paper, "cells were analyzed 72 h after transfection".

- Similarly, about the localization of the different isoforms, it would be interesting to know if the isoforms also localize to centrioles/centrosomes, can the authors provide these quantifications? Similarly for the interactors, would overexpression of type 3 for more than 48h prevent POC5 from localizing to the centriole?

Answer: Unfortunately, knockdown of FAM161A using RNAi in U20S cells have only a small effect on POC5 localization to the centriole even if "cells were analyzed 72 h after transfection". Please see "While FAM161A was absent in 88% of centrioles in FAM161A-depleted cells, only 38% of centrioles had lost POC5 (S8 Fig), suggesting that FAM161A does not control POC5 recruitment at centrioles" in Mercey, Olivier, et al. "The connecting cilium inner scaffold provides a structural foundation that protects against retinal degeneration." *Plos Biology* 20.6 (2022): e3001649. In this paper

Also, we find that type 3 mouse Fam161a localizes to the centriole probably because POC5 interacts with other centriolar proteins such as POC1B and others. However, we overexpressed both POC5 and the 3 isoforms of FAM161A individually, and we found that overexpressed Mice

FAM161A type 3 has lower colocalization rate with overexpressed POC5 outside the centrosomes (Fig 4D–F).

And Wouldn't a mouse cell culture model be more adequate to overexpress mouse proteins?

Answer: Thank you for suggesting these experiments we have done the following experiments in 3T3 cells:

- 1) House mouse and human FAM161A isoforms can localize to canonical centrioles in U2OS cells and 3T3 cells (SFig 4-1 and SFig 4-2)**
- 2) We tested the localization of expressing human and mouse FAM161A isoforms in 3T3 cells and found similar results to those observed in U2OS (SFig 4-4).**
- 3) Overexpressed Mice FAM161A type 3 has lower colocalization with overexpressed POC5 outside the centrosomes (Fig 4D-F).**

- In Figure 3B, it appears that there is a second band, potentially FAM161A type 2. Can the authors provide a more exposed blot?

Answer: More exposed blot of Fig3B has been provided (also, in SFig 3-2e for long exposure)

- Regarding the presence of isoforms, in addition to WBs, it would be important to demonstrate the presence of the protein isoforms using mass spectrometry analysis (with characteristic peptides of each form).

Answer: Thank you for this suggestion; however, Type 3 has only five unique amino acids not found in type 1 or 2. Therefore, finding this difference in mass spectrometry analysis would be highly challenging.

- The authors use an AlphaFold structure prediction to show positively selected sites and two apparent primary domains: core and outer. How confident is this model? The structure prediction of FAM161A is mostly unfolded, right? If the prediction does not have a high score, this type of modeling is misleading and incorrect.

Answer: Thank you for this comment; we have removed this prediction from the paper.

Reviewer #1 (Remarks to the Author):

The authors have made numerous revisions in response to the reviewers' comments. These have largely addressed the reviewers' concerns and have strengthened what was already a valuable contribution.

Reviewer #2 (Remarks to the Author):

After carefully reading the revised manuscript and the rebuttal letter, I fully support the publication of this work. My main concern that correlation was conflated with causation has been address.

Reviewer #3 (Remarks to the Author):

On the whole, the authors have improved the article and answered some of the questions I had. I therefore recommend the article for publication.

I think a minor correction needs to be made, the term "luminal scaffold" needs to be changed to "inner scaffold" to remain consistent with the literature.

Reviewer #4 (Remarks to the Author):

This is an excellent article that explains a great deal of biology, from an evolutionary transition in centriole structure to a candidate protein underlying that transition to a mechanism of that protein's important role in centriole degradation. It is well-written and organized, compelling, and large in scope.

As mentioned in a previous review, the authors have three papers rolled into one, a minireview on sperm morphology in Muridae, an evolutionary analysis of centrosomal proteins with special focus on FAM161A, and an experimental analysis of the role of FAM16A in spermatogenesis. I think rolling these three papers into one works very well, they build directly on one another and tell an interesting story.

The authors show strong evidence that 1) loss of sperm neck centrioles is a derived feature of Muridae 2) FAM161A and in particular type 3 are interesting candidate proteins worthy of functional tests with respect to centriole degradation and 3) house mouse type 2 and type 3 isoforms function differently from the human type 2 isoform in manner that is consistent with their playing an important role in centriole degradation. The images and the conclusions drawn from the data presented in Figures 4, 6 and 7 are strong. In addition, they show evidence of the gradual evolution from basal to derived sperm neck construction in close relatives to the Muridae (Figure 5) that is compelling and worth further investigation.

I think the work is very significant, first in substantiating the nascent claim that degenerate centrioles are a feature of Muridae and then in identifying a novel FAM161A type 3 isoform that has a dominant negative function relevant to the evolution of degenerate centrioles. There are few stories in evolution where a major evolutionary event can be somewhat tied to the evolution of a particular protein, and this is one such story.

My only suggestion to the authors would be to reduce the emphasis of embedding their work in cascade theory. Cascade theory is not testable and is not helpful in understanding their work, which stands very well on its own.

Additional comments:

Line 122-123 "Conservation on the one hand and diversity on the other produces genetic conflict and requires genomic plasticity, which should have a molecular signature of positive selection."

This statement refers to centriole biodiversity over very long time scales, it does not follow that genetic conflict is true for any particular lineage, let alone for every organism using centrioles. The dN/dS results show the centriole proteins experience purifying selection in mouse/basal mammal comparisons, including FAM161A - one would expect to find positive selection here, if anywhere.

Line 287 refers to STable 2-2 titled "In rodents, natural selective pressure on FAM161A is stronger than on other rod proteins and is strongest in Muridae." However, FAM161A is evolving more quickly than the other rod proteins. I don't think it follows that "natural selective pressure" is therefore stronger, if anything it would be weaker (less purifying).

Line 324 - "This finding is consistent with a weaker selective constraint on FAM161A primary structure in Muridae." This seems contradictory to the title of STable 2-2. I think the meaning of conserved and variable residues can be argued in both directions, so far as adaptation and constraint are concerned, and so should not be given too much emphasis.

Line 695 - "In the future, it will be important to identify the specific selective pressures that brought about mammalian centriole remodeling and centriole dispensability in some rodents." How would you do this? Can you suggest experiments that could address this point?

In addition, I would like to hear a little more about FAM161A and centriole development/degradation in the discussion and future directions:

I would be interested in discussion of the different engagements FAM161A mouse type 2 and 3 have with microtubules and centriole rod proteins, and what experiments/future directions the authors might pursue to explore these with respect to their role in centriole degradation. Related to this, I would be interested in experiments/future directions that would determine the basis of different localizations of mouse type 2 and type 3 forms compared to the human type 2 isoforms. Why does type 3 form foci in the nucleus?

I would be interested in experiments/future directions investigating the dominant negative behavior of type 3. Is it the small exon 5 motif? Do exons 4, 6, and 7 play a role in FAM161A engagement with microtubules or rod proteins?

I would be interested in CRISPER/CAS experiments that replace the mouse type 2 and type 3 isoforms the human type 2 isoform in the germline, to see if the ancestral sperm neck structure is generated in part or in whole. The ability of FAM161A mouse types 2 and 3 to function in the same manner in U2OS and UT3 cells suggests there is little functional evolution on the part of the other centriole proteins suggesting the FAM161A is largely determinative of centriole development in the sperm neck.

I would be interested in more discussion of the results in Figure 5, which indicate a gradual progression from ancestral to degenerate centrioles in the evolution of Muridae from a shared ancestor with Cricetidae. Similar experiments to those in this article using FAM161A and POC1 isoforms from house mouse, deer mouse and prairie vole would be interesting, is there a hierarchy in the order of evolution in these proteins?

A point-by-point response to these specific editorial requests

Reviewer #1 (Remarks to the Author):

The authors have made numerous revisions in response to the reviewers' comments. These have largely addressed the reviewers' concerns and have strengthened what was already a valuable contribution.

Answer: Thank you

Reviewer #2 (Remarks to the Author):

After carefully reading the revised manuscript and the rebuttal letter, I fully support the publication of this work. My main concern that correlation was conflated with causation has been address.

Answer: Thank you

Reviewer #3 (Remarks to the Author):

On the whole, the authors have improved the article and answered some of the questions I had. I therefore recommend the article for publication.

Answer: Thank you

I think a minor correction needs to be made, the term "luminal scaffold" needs to be changed to "inner scaffold" to remain consistent with the literature.

Answer: Corrected

Reviewer #4 (Remarks to the Author):

This is an excellent article that explains a great deal of biology, from an evolutionary transition in centriole structure to a candidate protein underlying that transition to a mechanism of that protein's important role in centriole degradation. It is well-written and organized, compelling, and large in scope.

As mentioned in a previous review, the authors have three papers rolled into one, a minireview on sperm morphology in Muridae, an evolutionary analysis of centrosomal proteins with a special focus on FAM161A, and an experimental analysis of the role of FAM16A in spermatogenesis. I think rolling these three papers into one works very well, they build directly on one another and tell an interesting story.

The authors show strong evidence that

- 1) loss of sperm neck centrioles is a derived feature of Muridae
- 2) FAM161A and in particular type 3 are interesting candidate proteins worthy of functional tests with respect to centriole degradation and
- 3) house mouse type 2 and type 3 isoforms function differently from the human type 2 isoform in manner that is consistent with their playing an important role in centriole degradation.

The images and the conclusions drawn from the data presented in Figures 4, 6 and 7 are strong. In addition, they show evidence of the gradual evolution from basal to derived sperm neck construction in close relatives to the Muridae (Figure 5) that is compelling and worth further investigation.

I think the work is very significant, first in substantiating the nascent claim that degenerate centrioles are a feature of Muridae and then in identifying a novel FAM161A type 3 isoform that has a dominant negative function relevant to the evolution of degenerate centrioles. There are few stories in evolution where a major evolutionary event can be somewhat tied to the evolution of a particular protein, and this is one such story.

My only suggestion to the authors would be to reduce the emphasis of embedding their work in cascade theory. Cascade theory is not testable and is not helpful in understanding their work, which stands very well on its own.

Answer: We appreciate the reviewer's excellent summary of the paper, which we have now incorporated into the discussion to address the points they raised below. Also, as suggested, we have reduced the emphasis on cascade theory in work by adding that the "Cascade theory is not testable."

Additional comments:

Line 122-123 "Conservation on the one hand and diversity on the other produces genetic conflict and requires genomic plasticity, which should have a molecular signature of positive selection."

This statement refers to centriole biodiversity over very long time scales, it does not follow that genetic conflict is true for any particular lineage, let alone for every organism using centrioles. The dN/dS results show the centriole proteins experience purifying selection in mouse/basal mammal comparisons, including FAM161A - one would expect to find positive selection here, if anywhere.

Answer:

We appreciate the reviewer's point. As suggested, we have modified this section in the text, adding that the changes are expected only "in some lineages" and discussing gene duplication and new protein isoforms instead of positive selection.

Line 287 refers to STable 2-2 titled "In rodents, natural selective pressure on FAM161A is stronger than on other rod proteins and is strongest in Muridae." However, FAM161A is evolving more quickly than the other rod proteins. I don't think it follows that "natural selective pressure" is therefore stronger, if anything it would be weaker (less purifying).

Answer: We appreciate the reviewer's point. As suggested, we changed this title to "In rodents, FAM161A is evolving more quickly than the other rod proteins in Muridae".

Line 324 – "This finding is consistent with a weaker selective constraint on FAM161A primary structure in Muridae." This seems contradictory to the title of STable 2-2. I think the meaning of conserved and variable residues can be argued in both directions, so far as adaptation and constraint are concerned, and so should not be given too much emphasis.

Answer: We appreciate the reviewer's point. As suggested, to de-emphasize this point, we deleted this sentence.

Line 695 - "In the future, it will be important to identify the specific selective pressures that brought about mammalian centriole remodeling and centriole dispensability in some rodents." How would you do this? Can you suggest experiments that could address this point?

Answer: We appreciate the reviewer's point. As suggested, we added this information to the discussion section.

In addition, I would like to hear a little more about FAM161A and centriole development/degradation in the discussion and future directions:

I would be interested in discussion of the different engagements FAM161A mouse type 2 and 3 have with microtubules and centriole rod proteins, and what experiments/future directions the authors might pursue to explore these with respect to their role in centriole degradation.

Related to this, I would be interested in experiments/future directions that would determine the basis of different localizations of mouse type 2 and type 3 forms compared to the human type 2 isoforms. Why does type 3 form foci in the nucleus?

Answer: We appreciate the reviewer's point. As suggested, we added this information to the discussion section.

I would be interested in experiments/future directions investigating the dominant negative behavior of type 3. Is it the small exon 5 motif? Do exons 4, 6, and 7 play a role in FAM161A engagement with microtubules or rod proteins?

Answer: We appreciate the reviewer's point. As suggested, we added this information to the discussion section.

I would be interested in CRISPER/CAS experiments that replace the mouse type 2 and type 3 isoforms the human type 2 isoform in the germline, to see if the ancestral sperm neck structure is generated in part or in whole. The ability of FAM161A mouse types 2 and 3 to function in the same manner in U2OS and UT3 cells suggests there is little functional evolution on the part of the other centriole proteins suggesting the FAM161A is largely determinative of centriole development in the sperm neck.

Answer: We appreciate the reviewer's point. As suggested, we added this information to the discussion section.

I would be interested in more discussion of the results in Figure 5, which indicate a gradual progression from ancestral to degenerate centrioles in the evolution of Muridae from a shared ancestor with Cricetidae. Similar experiments to those in this article using FAM161A and POC1 isoforms from house mouse, deer mouse and prairie vole would be interesting, is there a hierarchy in the order of evolution in these proteins?

Answer: We appreciate the reviewer's point. As suggested, we added this information to the discussion section.